# EFFICIENT OFFLINE REINFORCEMENT LEARNING VIA PEER-INFLUENCED CONSTRAINT

**Yujia Zhang**[1], **Lin Li**[2], **Wei Wei**[2*], **Jianguo Wu**[2], **Yi Ma**[2], **Jiye Liang**[2]

[1] School of Computer Science and Technology, North University of China,
Taiyuan, 030051, Shanxi, China

[2] Key Laboratory of Computational Intelligence and Chinese Information Processing of Ministry
of Education, School of Computer and Information Technology, Shanxi University,
Taiyuan, 030006, Shanxi, China

E-mails: `342564535@qq.com`, `zhangyj@nuc.edu.cn` (Yujia Zhang);
`weiwei@sxu.edu.cn` (Wei Wei, corresponding author)

## ABSTRACT

Offline reinforcement learning (RL) seeks to learn an optimal policy from a fixed
dataset, but distributional shift between the dataset and the learned policy often
leads to suboptimal real-world performance. Existing methods typically use be-
havior policy regularization to constrain the learned policy, but these conservative
approaches can limit performance and generalization, especially when the behav-
ior policy is suboptimal. We propose a Peer-Influenced Constraint (PIC) frame-
work with a "peer review" mechanism. Specifically, we construct a set of similar
states and use the corresponding actions as candidates, from which we select the
optimal action to constrain the policy. This method helps the policy escape local
optima while approximately ensuring the staying within the in-distribution space,
boosting both performance and generalization. We also introduce an improved
version, Ensemble Peer-Influenced Constraint (EPIC), which combines ensemble
methods to achieve strong performance while maintaining high efficiency. Addi-
tionally, we uncover the Coupling Effect between PIC and uncertainty estimation,
providing valuable insights for offline RL. We evaluate our methods on classic
continuous control tasks from the D4RL benchmark, with both PIC and EPIC
achieving competitive performance compared to state-of-the-art approaches.

## 1 INTRODUCTION

Reinforcement learning (RL) focuses on optimizing sequential decision-making in dynamic envi-
ronments (Sutton & Barto, 2018). By leveraging neural networks, online RL has achieved success
in gaming and robotic manipulation (Mnih et al., 2013; 2015; Zhang et al., 2024a; Shi et al., 2024).
However, its reliance on continuous interaction with the environment limits its application in sce-
narios where data collection is expensive, time-consuming, or dangerous (Cao et al., 2025; Yu et al.,
2021a). Offline RL addresses this by learning from a fixed, pre-collected dataset, eliminating the
need for further interactions during training (Lange et al., 2012; Levine et al., 2020). Although
promising, traditional online RL algorithms often struggle in offline settings due to the lack of on-
line interaction for error correction, leading to instability and potential learning failures when errors
propagate (Fujimoto et al., 2019; Tarasov et al., 2023a; Zhang et al., 2024b; 2025a). This difficulty
in accurately evaluating out-of-distribution (OOD) actions further challenges the value-based policy
learning process.

To address distributional shifts between the dataset and the learned policy in offline RL, regular-
ization is often applied during training to encourage policies to stay close to dataset actions during
evaluation and improvement. Existing methods can be broadly categorized into value regularization
and policy regularization methods. Value regularization methods penalize the Q-values of OOD
actions to discourage them from being selected (Kumar et al., 2020; Yu et al., 2021b; Wu et al.,
2021; Lyu et al., 2022; Nakamoto et al., 2024). Methods such as SAC-N and EDAC (An et al.,
2021) use ensembles to estimate value distributions and apply pessimistic penalties. However, these
methods often require large ensembles, leading to high computational costs and slower convergence

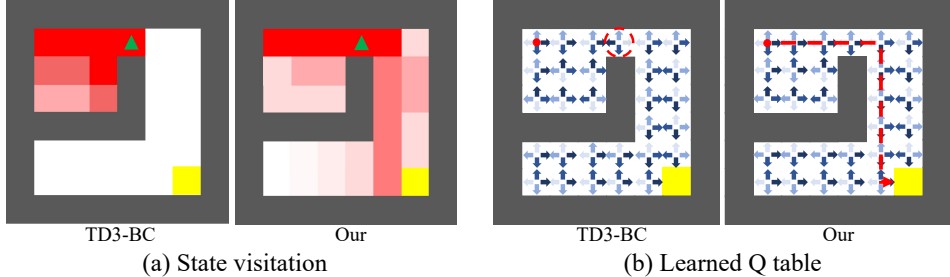

|              |      |              |     |
|--------------|------|--------------|-----|
| TD3-BC       | Our  | TD3-BC       | Our |
| (a) State visitation |  | (b) Learned Q table |  |

Figure 1: The illustrations and results of the motivation experiment. (a) State visitation heatmap (darker = higher probability): PIC escapes local optima and reaches the goal, while TD3+BC, constrained by its one-to-one state–action mapping, stalls. (b) Q-table visualization (darker arrows = higher $Q$): PIC drives the agent toward the goal; TD3+BC remains pessimistic.

speed. Additionally, many uncertainty estimators overlook critical information, such as inter-sample distances, which are essential for distinguishing in-distribution and OOD actions in RL.

Policy regularization methods constrain the policy by focusing on actions within the dataset, thereby avoiding OOD actions without relying on ensemble modeling (Kostrikov et al., 2021a; Nair et al., 2020; Fujimoto & Gu, 2021; Ma et al., 2024; Luo et al., 2024). While these methods are computationally efficient, their generalization is limited by the restricted set of available actions, which may not always be optimal. For example, TD3+BC (Fujimoto & Gu, 2021) shows limited generalization in a 2D grid world task (Sutton & Barto, 2018; Xu et al., 2023b) when the dataset contains missing actions, such as moving right (illustrated by the green triangle in Figure 1, see Appendix E.1 for details). This raises a critical question: *How can we more effectively utilize dataset information to enhance generalization while preserving computational efficiency?*

To answer the above question, we propose Peer-Influenced Constraint (PIC), a plug-in regularizer that exploits relationships within the dataset via a "peer review" mechanism. For a query state, PIC reuses actions from similar peer states and steers the policy toward a high-value candidate. This expands the effective in-dataset action set—reusing actions observed elsewhere rather than relying only on the behavior action—while remaining compute-light and ensemble-agnostic. When combined with ensembles, we observe a Coupling Effect between PIC strength and uncertainty estimation, enabling effective OOD penalties with substantially smaller ensembles. Building on this, we develop Ensemble Peer-Influenced Constraint (EPIC) to further improve data-coverage generalization at competitive training cost.

The contributions of this paper are as follows:

- We introduce PIC, a drop-in policy regularization module that reuses cross-state information in the dataset to guide the policy toward high-value, in-distribution actions, without training additional generative models or relying on large ensembles.

- We identify a Coupling Effect between PIC strength and uncertainty estimation that enables effective OOD penalties with smaller ensembles; building on this, we introduce EPIC, a PIC-enhanced ensemble with strong performance and high efficiency.

- We instantiate PIC in TD3, SAC, and IQL, demonstrating competitive performance and computational efficiency on offline RL benchmarks, as well as in offline-to-online and controlled generalization evaluations.

## 2 PRELIMINARIES

In this section, we briefly introduce the common RL settings, notations, and details of offline RL methods.

## 2.1 REINFORCEMENT LEARNING

In RL, the environment is typically modeled as an infinite-horizon Markov Decision Process (MDP), represented by the tuple $(\mathcal{S}, \mathcal{A}, \mathcal{P}, p_0, r, \gamma)$. This includes a state space $\mathcal{S}$, an action space $\mathcal{A}$, transition probabilities $\mathcal{P} : \mathcal{S} \times \mathcal{A} \times \mathcal{S} \rightarrow [0, 1]$, an initial state distribution $p_0$, a reward function $r : \mathcal{S} \times \mathcal{A} \times \mathcal{S} \rightarrow \mathbb{R}$, and a discount factor $\gamma \in [0, 1]$.

Given the MDP and the agent's policy $\pi_\phi$ with parameter $\phi$, at time $t$, the agent observes the state $s_t$ and selects an action $a_t = \pi_\phi(s_t)$. The environment then provides a reward $r(s_t, a_t)$ and transitions to the next state $s_{t+1}$. Let $R_t = \sum_{t=0}^{\infty} \gamma^t r(s_t, a_t)$ denote the expected discounted reward of $\pi_\phi$ starting from time $t$, accounting for the randomness in the initial state distribution $p_0$ and the transition function $\mathcal{P}$. The Q-value $Q^\pi : \mathcal{S} \times \mathcal{A} \rightarrow \mathbb{R}$ represents the expected cumulative return obtained by taking action $a$ at state $s$ and subsequently following the policy $\pi_\phi$. It quantifies the long-term value of a state-action pair under the given policy:

$$Q^\pi(s, a) = \mathbb{E}\left[\sum_{t=0}^{T} \gamma^t r(s_t, a_t) | s_0 = s, a_0 = a\right]. \tag{1}$$

The goal in RL is to find the optimal policy $\pi^*(s) \in \text{argmax}_a Q(s, a)$, maximizing $R_t$.

## 2.2 OFFLINE REINFORCEMENT LEARNING

Classic RL requires environment interaction during training, which is impractical in tasks with high safety or cost constraints. Offline RL addresses this by learning an optimal policy from a pre-existing dataset $\mathcal{D} = (s, a, r, s')$ without environment access. However, adapting existing algorithms often leads to suboptimal policies due to overestimation bias for unseen state-action pairs, which is corrected in online RL through exploration and feedback but requires additional mechanisms in offline RL to stabilize training.

To counteract overestimation bias, a common approach is to address the root cause: the Q-value estimates. Uncertainty estimation helps by highlighting that Q-values for OOD actions are more uncertain than those for in-sample actions. Both SAC-N and EDAC (An et al., 2021) use an ensemble of $N$ Q-functions to approximate the Q-value distribution. They update the $i$-th critic parameter $\theta_i$ by minimizing the squared temporal difference loss:

$$L(\theta_i) = \mathbb{E}_{(s,a,r,s') \sim \mathcal{B}}\left[(y - Q_{\theta_i}(s, a))^2\right], \tag{2}$$

where $y = r + \gamma \min_{i=1,\dots,N} Q_{\theta_i'}(s', \pi_{\phi'}(s'))$, $\theta'$ and $\phi'$ represent the parameters of the target networks for the critic and actor, respectively, $\mathcal{B}$ is a minibatch from $\mathcal{D}$.

The policy can be improved by:

$$L_1(\phi) = \mathbb{E}_{s \sim \mathcal{B}}\left[-\min_{i=1,\dots,N} Q_{\theta_i}(s, \pi_\phi(s))\right]. \tag{3}$$

Using the minimum Q-value penalizes OOD actions and avoids overestimation with large $N$, but methods such as SAC-N and EDAC incur high cost from maintaining $N$ Q-networks.

In addition to value regularization methods, policy constraint methods directly constrain the learned policy to stay close to the behavior policy in the dataset, thereby avoiding OOD actions. For example, TD3+BC (Fujimoto & Gu, 2021) extends the Equation (3) by incorporating a behavior cloning loss term into the policy update:

$$L_{\text{BC}}(\phi) = \mathbb{E}_{(s,a) \sim \mathcal{B}}\left[||a - \pi_\phi(s)||^2\right]. \tag{4}$$

TD3+BC achieves competitive performance with lower computational costs compared to ensemble-based methods. However, it often struggles in datasets with limited action diversity, which restricts policy improvement and leads to suboptimal generalization.

## 3 METHOD

In this section, we introduce our method. We first define PIC and establish performance gap bounds, before integrating PIC into an offline RL algorithm. We then extend PIC to ensemble-based RL

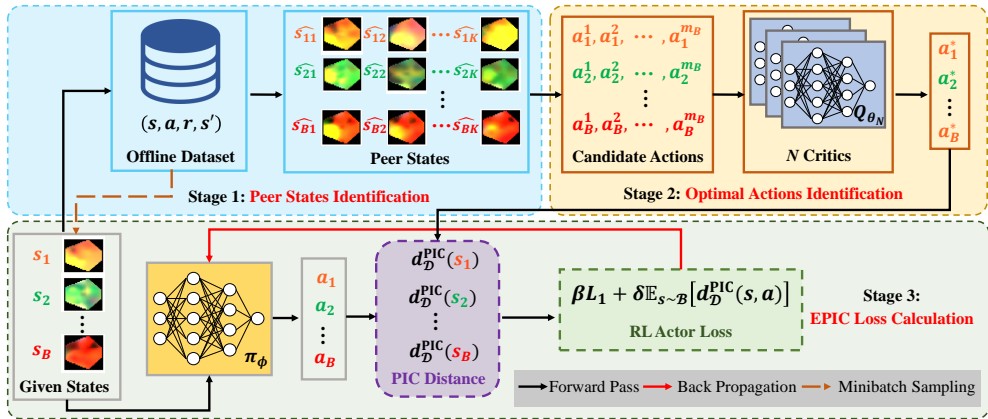

Figure 2: Actor training workflow of EPIC. It consists of three stages: identifying peer states, selecting optimal actions, and computing the PIC loss. First, peer states are retrieved for a given state. Next, the corresponding optimal actions are determined. Finally, the PIC distance is calculated, and the actor is updated by jointly optimizing the Q-value loss $L_1$ and the PIC loss.

methods, where we uncover a Coupling Effect between PIC and uncertainty quantification. Building on these insights, we propose EPIC, which unifies PIC and the Coupling Effect to enable efficient and generalized policy learning. The gradient propagation of actor training in EPIC is illustrated in Figure 2.

## 3.1 PEER-INFLUENCED CONSTRAINT

Inspired by representation learning-based RL methods (Fujimoto et al., 2023; Ni et al., 2024) and dataset-constrained approaches (Ran et al., 2023), which capture structural information in the environment by mapping similar states or state-action pairs to similar embeddings, we observe that offline RL datasets often contain rich but underutilized structural information about states. Specifically, we propose to utilize the inherent structure of the dataset, where states with similar characteristics (e.g., proximity or shared features) often contain valuable information for guiding policy learning. We first introduce PIC, a novel approach that leverages peer states to enhance policy learning in offline RL. We then describe an efficient peer states search method for fast and effective state matching. Finally, we present a practical algorithm that integrates PIC into TD3 to enhance offline RL performance.

**Definition 1** (PIC Distance). *Let $s$ be a state from the dataset $\mathcal{D}$, and let $\pi_\phi(s)$ be a parameterized policy. The PIC distance is defined as:*

$$d_{\mathcal{D}}^{PIC}(s) = \|\pi_\phi(s) - a^*\|, \tag{5}$$

*where $a^*$ is the optimal action selected from a candidate action set $\mathcal{A}'$.*

The optimal action $a^*$ is determined as:

$$a^* = \arg\max_{a \in \mathcal{A}'} \min_{i=1,2} Q_{\theta_i}(s, a), \tag{6}$$

where $Q_{\theta_1}$ and $Q_{\theta_2}$ are the Q-values estimated by the two critics. To address potential overestimation issues in critics, we use two critics to provide more reliable candidate action value estimates (Fujimoto & Gu, 2021; Zhang et al., 2025b).

The candidate set $\mathcal{A}'$ includes: (1) Actions of the given state $s$; (2) Actions corresponding to the $K$ peer states. Each peer state, $\hat{s}_j$, is determined as: To ensure that peer states are distinct from $s$, we explicitly exclude $s$ during nearest-neighbor retrieval. Formally, the $j$-th peer state $\hat{s}_j$ is defined as:

$$\hat{s}_j = \arg\min_{\hat{s} \in \mathcal{D} \setminus (\mathcal{D}_{j-1} \cup \{s\})} \|s - \hat{s}\|, \qquad j = 1, 2, \dots, K, \tag{7}$$

where $\mathcal{D}_{j-1}$ is the set of already selected $j-1$ peer states of $s$. Since one state in $\mathcal{D}$ could correspond to multiple actions, the size of $\mathcal{A}'$ may be greater than $K + 1$.

PIC is built upon the assumption that, in continuous control domains with local smoothness, nearby states tend to share similar optimal actions, which makes the peer-based action constraint both feasible and effective. This approach yields a more optimal behavior constraint by accounting for actions of the given state, potential peer-state actions, effectively escaping local optima for more generalized policy learning. Furthermore, we provide an analysis of the performance gap between the learned policy $\pi_\phi$ using PIC constraint and the optimal policy $\pi^*$.

**Theorem 1** (Performance Gap of PIC). *Assume that $\mu$ and $P(s'|s,a)$ are Lipschitz continuous. For any $(s,a)$ and its peer pair $(\hat{s}_j, \hat{a}_j)$, suppose $||\hat{s}_j - s|| \leq \epsilon_s$, $||\mu(\hat{s}_j) - \pi(s)|| \leq \epsilon_a$, and $\max_{s \in S} |\pi^*(s) - \mu(s)| \leq \epsilon_*$, conditions that can be satisfied by PIC. Then, the following inequality holds:*

$$|J(\pi^*) - J(\pi)| \leq \frac{CL_P R_{\max}}{1 - \gamma}(\epsilon_a + L_s \epsilon_s + \epsilon_*), \tag{8}$$

*where $C$ is a positive constant, $L_P$ and $L_s$ represent the Lipschitz constant, $R_{\max}$ is the max value of reward.*

For a detailed proof, please see Appendix B. Observed from Theorem 1, the performance gap is influenced by $\epsilon_a$, $\epsilon_s$, and $\epsilon_*$. PIC reduces $\epsilon_s$ by ensuring that peer states $\hat{s}_j$ are close to the given state $s$, thereby minimizing state deviations. PIC reduces $\epsilon_a$ by selecting actions $a^*$ from the candidate set that are close to both the behavior policy $\mu(s)$ and the optimal policy $\pi^*(s)$, thus minimizing the action discrepancy. Finally, PIC reduces $\epsilon_*$ by aligning the learned policy with the optimal policy, narrowing the performance gap and improving generalization.

PIC is similar to TD3+BC (Fujimoto & Gu, 2021), but with significant differences. While TD3+BC relies on strict state-action correspondences, limiting exploration and adaptability, our approach avoids these constraints. Unlike methods that minimize the distance between $(s, \pi_\phi(s))$ and its nearest state-action pair in $\mathcal{D}$ (Ran et al., 2023), which can result in incorrect constraints due to state mismatches, our method considers actions from both the given state and similar states. This enables optimal action selection without state mismatches and eliminates the need for extensive tuning, ensuring better performance, avoiding the OOD issues.

### 3.1.1 Efficient Peer States Search

PIC identifies, for each state $s \in \mathcal{D}$, its $K$ nearest peer states in the state space. To accelerate retrieval, we adopt a precomputed KD-Tree (Bentley, 1975), a standard index for efficient nearest-neighbor search (Ran et al., 2023; Zhang et al., 2023), which significantly reduces training-time computation by avoiding redundant searches. Concretely, we build the KD-Tree over all dataset states once before training, and query it during training to fetch the $K$ nearest states in $\mathcal{O}(|s| \log |\mathcal{D}|)$ time per query (Ram & Sinha, 2019), where $|s|$ denotes the state dimensionality. Unlike dataset-constraint methods (Ran et al., 2023) that search jointly over state and action spaces, our approach compares states only, lowering computational cost while maintaining accuracy.

### 3.1.2 Practical Algorithm

PIC is versatile and can be integrated as a component into any Actor-Critic methods. In this paper, we combine it with the straightforward and easy-to-implement TD3 method, introducing a practical algorithm called PIC-TD3. The actor update for PIC-TD3 is:

$$L_{\mathrm{PT}}(\phi) = \mathbb{E}_{s \sim \mathcal{B}}\left[-\beta Q_{\theta_1}(s, a)\right] + \delta \mathbb{E}_{s \sim \mathcal{B}}\left[d_{\mathcal{D}}^{\mathrm{PIC}}(s)\right], \tag{9}$$

where $a = \pi_\phi(s)$, $\beta = \frac{\alpha|\mathcal{B}|}{\sum_{s_i, a_i} Q(s_i, a_i)}$, and the $\alpha$ in $\beta$ is a hyper-parameter. $\delta$ is the PIC strength parameter. The setting of $\beta$ is similar to that in TD3+BC (Fujimoto & Gu, 2021), designed to mitigate sensitivity to Q-value estimates. The pseudocode of PIC-TD3 is provided in Appendix C.

Note that PIC is designed as a versatile plugin that can be seamlessly integrated into different RL algorithms without requiring major architectural modifications. In Appendix F.3, we incorporate PIC into IQL (Kostrikov et al., 2021b) and SAC (Haarnoja et al., 2018), showing that it consistently improves performance and scales effectively.

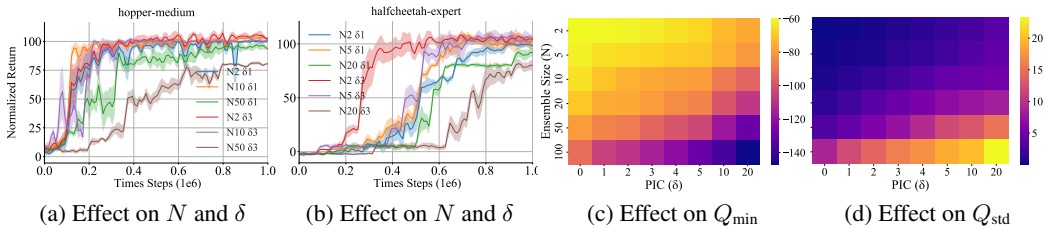

Figure 3: The impact of ensemble size $N$ and PIC strength $\delta$ on performance and uncertainty.

## 3.2 ENSEMBLE PEER-INFLUENCED CONSTRAINT

In this section, we first assess how PIC augments ensemble-based methods and reveal a Coupling Effect between PIC strength and uncertainty estimation. We then introduce EPIC, which attains state-of-the-art performance with strong generalization and improved computational efficiency. Finally, we discuss why EPIC works.

### 3.2.1 INTEGRATING PIC WITH ENSEMBLE METHODS

Policy constraint methods are typically more efficient and stable, but ensemble-based value constraint methods, like EDAC (An et al., 2021), can perform well in certain environments. However, these methods often require large networks and extensive resources, especially on some expert-level datasets. We explore whether PIC can bring benefits to such ensemble methods by integrating it with EDAC and testing its effectiveness. As shown in Figure 3 (a) and (b), decent performance is achieved with $N = 2$ (the original PIC-TD3 version), and performance improves with increasing $N$. However, when $N$ is too large, such as in hopper-medium ($N = 50$) or halfcheetah-expert ($N = 20$), the policy converges more slowly, and the convergence performance also degrades. Additionally, we find that higher PIC strength reduces the number of critics needed to achieve comparable performance.

We further measure and analyze the relationship between various uncertainty metrics, ensemble size $N$, and PIC strength $\delta$ on the hopper-medium-expert dataset. We sample 10,000 state-action pairs and compute the uncertainty metrics $Q_{\min}$, $Q_{\text{std}}$, and $Q_{\text{clip}} = Q_{\text{mean}} - Q_{\min}$ at the same time step under different values of $N$ and $\delta$. We then calculate the average values and present the heatmaps in Figure 3 (c) and (d); results for $Q_{\text{clip}}$ are reported in Appendix F.1.

Our findings show that, for a fixed $N$, increasing $\delta$ raises uncertainty, allowing smaller ensembles to maintain sufficient penalties for OOD actions. Conversely, for a fixed $\delta$, increasing $N$ results in higher uncertainty as well. We analyze that as PIC strength increases, the policy increasingly selects in-distribution actions while assigning more conservative values to potential OOD actions. As a consequence, $Q_{\min}$ becomes more pessimistic, thereby further reducing overestimation, and ensemble disagreement $Q_{\text{std}}$ tends to increase on OOD candidates, and $Q_{\text{clip}}$ also tends to increase. These changes reflect a stronger penalization of high-risk, potentially OOD actions, similar to the increased OOD penalty produced by using a larger ensemble size.

These observations suggest that without PIC, Q-value estimates for OOD actions often exhibit low uncertainty, leading to the overestimation of their values. This in turn requires the use of large ensembles to mitigate the risk of inaccurate Q-value predictions and ensure reliable performance. By incorporating PIC, we introduce a "Coupling Effect" that not only improves policy performance with smaller ensembles but also significantly reduces computational overhead. Furthermore, we measure the fraction of policy actions that lie in in-distribution regions under different training steps, PIC strengths $\delta$, and ensemble sizes $N$. The results, reported in Appendix F.1, show that introducing PIC significantly increases the in-distribution ratio compared to the no-PIC baseline. Moreover, this ratio consistently grows as $\delta$, the training horizon, and $N$ increase, indicating that the policy is progressively concentrated on the dataset support. From the perspective of the policy distribution, this provides additional evidence for the coupling effect: stronger PIC constraints and larger ensembles jointly help keep the learned policy close to in-distribution regions.

---

**Algorithm 1** Ensemble Peer-Influenced Constraint Algorithm (EPIC)

---

**Input:** Observation state $\mathcal{S}$, offline dataset $\mathcal{D}$
**Output:** Learned policy $\pi$
Initialize: Critic networks $\{Q_{\theta_i}\}_{i=1}^N$, actor network $\pi_\phi$, target networks $\{Q_{\theta_i'}\}_{i=1}^N$, $\phi'$, and peer state number $K$
**for** $t = 1$ to $T$ **do**
    Sample a mini-batch $\mathcal{B} = \{(s, a, r, s')\}$ from $\mathcal{D}$
    Find peer states in $\mathcal{D}$ of sample state by $\hat{s}_j = \arg\min_{\hat{s} \in \mathcal{D} \setminus \mathcal{D}_{j-1}} \|s - \hat{s}\|$
    Construct candidate action set $\mathcal{A}'$
    Select the optimal action $a^*$ using $a^* = \arg\max_{a \in \mathcal{A}'} \min_{i=1,2,\cdots,N} Q_{\theta_i}(s, a)$
    Calculate PIC distance of sample state by $d_{\mathcal{D}}^{\text{PIC}}(s) = \|\pi_\phi(s) - a^*\|$
    Update $\theta_i$ by minimizing $L_{\text{EPIC}}(\theta_i) = \mathbb{E}_{(s,a,r,s') \sim \mathcal{B}}\left[(y - Q_{\theta_i}(s,a))^2 + \text{ES}\right]$
    Update $\phi$ using gradient descent with $L_{\text{EPIC}}(\phi) = \beta L_1(\phi) + \delta \mathbb{E}_{s \sim \mathcal{B}}\left[d_{\mathcal{D}}^{\text{PIC}}(s)\right]$
    $\theta_i' \leftarrow \tau\theta_i + (1-\tau)\theta_i'$, $\phi' \leftarrow \tau\phi + (1-\tau)\phi'$
**end for**

---

Building on this Coupling Effect, we apply PIC to EDAC and introduce the EPIC, which produces state-of-the-art, generalized policies with high computational efficiency. The actor update for EPIC is as follows:

$$L_{\text{EPIC}}(\phi) = \beta L_1(\phi) + \delta \mathbb{E}_{s \sim \mathcal{B}}\left[d_{\mathcal{D}}^{\text{PIC}}(s)\right], \qquad (10)$$

where $L_1$ is defined in Equation (3) and the optimal action $a^*$ in $d_{\mathcal{D}}^{\text{PIC}}(s)$ is

$$a^* = \arg\max_{a \in \mathcal{A}'} \min_{i=1,2,\cdots,N} Q_{\theta_i}(s, a). \qquad (11)$$

The critic update method in EPIC is consistent with EDAC, with the addition of an ensemble similarity (ES) term to Equation (2):

$$L_{\text{EPIC}}(\theta_i) = \mathbb{E}_{(s,a,r,s') \sim \mathcal{B}}\left[(y - Q_{\theta_i}(s,a))^2 + \text{ES}\right], \qquad (12)$$

where $\text{ES} = \frac{\eta}{N-1}\sum_{1 \le i \ne j \le N}\left\langle \nabla_a Q_{\theta_i}(s,a), \nabla_a Q_{\theta_j}(s,a)\right\rangle$, $\eta$ is a hyperparameter and $N$ is the ensemble size. The ES term encourages diversity among critics, but to better evaluate the effectiveness of PIC, we set $\eta = 1$ in most environments. EPIC's pseudocode is provided in Algorithm 1.

### 3.2.2 WHY EPIC WORKS

EPIC combines the strengths of PIC and ensemble methods to enhance policy learning by leveraging in-sample optimal action selection and the Coupling Effect. It constrains the policy through PIC by constructing a candidate action set from similar states, avoiding unreliable OOD actions and local optima. By utilizing the diversity of multiple critics, EPIC stabilizes value estimates and uses a maxmin-like approach to select the optimal action $a^*$, effectively reducing overestimation and improving action selection.

Unlike traditional policy constraint methods (e.g., TD3+BC), which often struggle to surpass the performance of the behavior policy, or value constraint methods, which may inadequately penalize OOD actions, EPIC achieves a better balance between generalization and conservatism. The Coupling Effect between PIC and uncertainty quantification ensures an optimal trade-off among generalization, conservatism, and computational efficiency, resulting in stronger overall performance.

## 4 EXPERIMENTS

In this section, we conduct several experiments to evaluate the empirical performance of the proposed method. Our goal is to answer five key questions: (1) Does our method outperform previous approaches on standard offline benchmarks? (2) Is our method efficient? (3) Can our method learn optimal policies from peer states to enhance generalization? (4) How do different ensemble sizes $N$ and PIC strengths $\delta$ affect EPIC's performance? (5) Is our method still effective in offline-to-online fine-tuning?

Table 1: Average normalized score over final evaluations across five seeds on Gym-MuJoCo tasks.

| Task Name | TD3+BC | IQL | CQL | SAC-RND | PRDC | SAC-N | EDAC | PIC-TD3 (ours) | EPIC (ours) |
|---|---|---|---|---|---|---|---|---|---|
| halfcheetah-random | 30.9 | 19.5 | **31.1** | 27.6 | 26.9 | 28.0 | 28.4 | 25.3 ± 2.1 | 28.9 ± 1.4 |
| halfcheetah-medium | 54.7 | 50.0 | 46.9 | 66.4 | 63.5 | 67.5 | 65.9 | 68.4 ± 2.4 | **68.9** ± 2.7 |
| halfcheetah-expert | 93.4 | 95.5 | 97.3 | 102.6 | – | 105.2 | 106.8 | 104.2 ± 3.2 | **107.9** ± 6.4 |
| halfcheetah-medium-expert | 89.1 | 92.7 | 95.0 | **108.1** | 94.5 | 107.1 | 106.3 | 99.8 ± 1.6 | 103.8 ± 3.8 |
| halfcheetah-medium-replay | 45.0 | 42.1 | 45.3 | 51.2 | 55.0 | 63.9 | 61.3 | 63.1 ± 0.9 | **64.8** ± 1.5 |
| halfcheetah-full-replay | 75.0 | 75.0 | 76.9 | 81.2 | – | 84.5 | 84.6 | 83.7 ± 1.1 | **87.3** ± 1.0 |
| hopper-random | 8.5 | 10.1 | 5.3 | 19.6 | 26.8 | **31.3** | 25.3 | 25.4 ± 3.2 | 27.9 ± 4.2 |
| hopper-medium | 60.9 | 65.2 | 61.9 | 91.1 | 100.3 | 100.3 | 101.6 | 100.7 ± 1.5 | **102.2** ± 0.5 |
| hopper-expert | 109.6 | 108.8 | 106.5 | 109.8 | – | 110.3 | 110.1 | 111.8 ± 2.1 | **112.2** ± 1.5 |
| hopper-medium-expert | 87.8 | 85.5 | 96.9 | 109.8 | 109.2 | 110.1 | 110.7 | 105.6 ± 1.1 | **112.3** ± 0.9 |
| hopper-medium-replay | 55.1 | 89.6 | 86.3 | 97.2 | 100.1 | 101.8 | **102.8** | 100.5 ± 1.3 | 102.0 ± 1.0 |
| hopper-full-replay | 97.9 | 104.4 | 101.9 | 107.4 | – | 102.9 | 105.4 | 105.8 ± 0.8 | **107.6** ± 1.6 |
| walker2d-random | 2.0 | 11.3 | 5.1 | 18.7 | 5.0 | 21.7 | 16.6 | 14.6 ± 4.1 | **22.4** ± 1.6 |
| walker2d-medium | 77.7 | 80.7 | 79.5 | 92.7 | 85.2 | 87.9 | 92.5 | 94.9 ± 2.2 | **95.9** ± 3.2 |
| walker2d-expert | 110.0 | 96.9 | 109.3 | 104.5 | – | 107.4 | 115.1 | 114.6 ± 1.5 | **117.7** ± 2.1 |
| walker2d-medium-expert | 110.4 | 112.1 | 109.1 | 104.6 | 111.2 | **116.7** | 114.7 | 114.4 ± 0.9 | 116.4 ± 1.7 |
| walker2d-medium-replay | 68.0 | 75.4 | 76.8 | 89.4 | 92.0 | 78.7 | 87.1 | **95.5** ± 2.4 | 95.2 ± 2.2 |
| walker2d-full-replay | 90.3 | 97.5 | 94.2 | 105.3 | – | 94.6 | 99.8 | 103.0 ± 1.9 | **107.4** ± 2.2 |
| **Average** | 70.4 | 72.9 | 73.7 | 82.6 | – | 84.4 | 85.2 | 85.1 | **87.8** |

Table 2: Average normalized score over the final evaluation and five seeds on the AntMaze tasks.

| Task Name | TD3+BC | IQL | CQL | SAC-RND | PRDC | MSG | SAC-BC-N | PIC-TD3 (ours) | EPIC (ours) |
|---|---|---|---|---|---|---|---|---|---|
| antmaze-umaze | 66.3 | 83.3 | 74.0 | 97.0 | **98.8** | 97.9 | 98.6 | 96.4 ± 3.5 | 98.6 ± 0.5 |
| antmaze-umaze-diverse | 53.8 | 70.6 | 84.0 | 66.0 | 90.0 | 79.3 | 91.2 | 91.1 ± 2.3 | **94.3** ± 4.2 |
| antmaze-medium-play | 26.5 | 64.6 | 61.2 | 38.5 | 82.8 | 85.9 | 85.8 | 79.3 ± 5.4 | **88.1** ± 3.3 |
| antmaze-medium-diverse | 25.9 | 61.7 | 53.7 | 74.7 | 78.8 | **84.6** | 73.8 | 77.9 ± 6.8 | 83.8 ± 4.0 |
| antmaze-large-play | 0.0 | 42.5 | 15.8 | 43.9 | 54.8 | 64.3 | 65.8 | 55.7 ± 7.7 | **65.9** ± 2.3 |
| antmaze-large-diverse | 0.0 | 27.6 | 14.9 | 45.7 | 50.0 | 71.3 | **75.8** | 53.2 ± 4.7 | 66.9 ± 5.3 |
| **Average** | 28.8 | 58.4 | 50.6 | 61.0 | 75.9 | 80.6 | 81.8 | 75.6 | **82.9** |

## 4.1 MAIN RESULTS ON BENCHMARKS

We evaluate the proposed method on three suites of D4RL tasks (Fu et al., 2020): Gym-MuJoCo, AntMaze, and Adroit. For each suite, we consider all available datasets. We compare our results to several ensemble-free baselines, including TD3+BC (Fujimoto & Gu, 2021), IQL (Kostrikov et al., 2021b), CQL (Kumar et al., 2020), SAC-RND (Nikulin et al., 2023), and PRDC (Ran et al., 2023). Given the effectiveness and efficiency of our approach, we also compare it to ensemble-based baselines, including SAC-N, EDAC (An et al., 2021), MSG (Ghasemipour et al., 2022) and SAC-BC-N (Beeson & Montana, 2024). Detailed implementation and baseline result sources are provided in Appendix D.

We train PIC-TD3 and EPIC for one million steps on each task within the Gym-MuJoCo, AntMaze, and Adroit benchmarks. The test results are presented in Tables 1, 2, and 3. In these tables, the mean-wise best results among algorithms are highlighted in bold, the second-best performance is underlined, and the symbol ± represents the standard deviation across the seeds. Our method, EPIC, outperforms all baselines in the vast majority of tasks, demonstrating the effectiveness of the PIC and Coupling Effect. Additionally, PIC-TD3 surpasses ensemble-free methods and achieves performance comparable to ensemble-based methods.

Appendix F.2 reports the minimum Q-ensembles for the results in Tables 1, 2, and 3, showing that our method achieves competitive performance with substantially fewer critics, thus highlighting its efficiency and robustness. Appendix F.8 provides additional training curves, which highlight the stability, convergence speed, and overall learning dynamics of our method. Appendix F.3 extends the PIC framework to other RL algorithms, where consistent gains further confirm its effectiveness in enhancing policy generalization and robustness.

Table 3: Average normalized score over the final evaluation and five seeds on the Adroit tasks.

| Task Name | BC | TD3+BC | IQL | CQL | SAC-RND | EDAC | RORL | PIC-TD3 (ours) | EPIC (ours) |
|---|---|---|---|---|---|---|---|---|---|
| pen-human | 34.4 | 81.8 | 81.5 | 37.5 | 5.6 | 51.2 | 33.7 | 84.7 $\pm$ 2.4 | **111.7** $\pm$ 5.5 |
| pen-cloned | 56.9 | 61.4 | 77.2 | 39.2 | 2.5 | 68.2 | 35.7 | 80.2 $\pm$ 1.1 | **94.6** $\pm$ 8.6 |
| pen-expert | 85.1 | 146.0 | 133.6 | 107.0 | 45.4 | 122.8 | 130.3 | 133.7 $\pm$ 4.7 | **150.9** $\pm$ 6.4 |
| door-human | 0.5 | -0.1 | 3.1 | 9.9 | 0.0 | 10.7 | 3.7 | 3.7 $\pm$ 1.3 | **12.9** $\pm$ 6.2 |
| door-cloned | -0.1 | 0.1 | 0.8 | 0.4 | 0.2 | **9.6** | -0.1 | 2.1 $\pm$ 0.8 | 7.1 $\pm$ 3.3 |
| door-expert | 34.9 | 84.6 | 105.3 | 101.5 | 73.6 | -0.3 | 104.9 | 105.6 $\pm$ 4.1 | **108.4** $\pm$ 1.4 |
| hammer-human | 1.5 | 0.4 | 2.5 | 4.4 | -0.1 | 0.8 | 2.3 | 4.9 $\pm$ 2.1 | **9.2** $\pm$ 6.5 |
| hammer-cloned | 0.8 | 0.8 | 1.1 | 2.1 | 0.1 | 0.3 | 1.7 | 2.4 $\pm$ 1.2 | **12.7** $\pm$ 3.3 |
| hammer-expert | 125.6 | 117.0 | 129.6 | 86.7 | 24.8 | 0.2 | **132.2** | 121.7 $\pm$ 3.1 | 130.4 $\pm$ 3.8 |
| relocate-human | 0.0 | -0.2 | 0.1 | 0.2 | 0.0 | 0.1 | 0.0 | 0.5 $\pm$ 0.7 | **2.7** $\pm$ 1.2 |
| relocate-cloned | -0.1 | -0.1 | 0.2 | -0.1 | 0.0 | 0.0 | 0.0 | 0.3 $\pm$ 0.2 | **1.1** $\pm$ 0.5 |
| relocate-expert | 101.3 | 107.3 | 106.5 | 95.0 | 3.4 | -0.3 | 47.8 | 105.7 $\pm$ 2.1 | **108.4** $\pm$ 1.4 |
| **Average** | 36.7 | 49.9 | 53.5 | 40.3 | 13.0 | 21.9 | 41.0 | 53.8 | **62.5** |

## 4.2 COMPUTATIONAL EFFICIENCY COMPARISON

Time complexity plays a critical role in offline RL. We evaluate PIC-TD3, EPIC, and several baseline methods on the hopper-medium dataset. Appendix F.4 shows training time (relative to TD3+BC) and normalized scores. Ensemble methods are accurate but costly, BC-based methods efficient but weaker, while EPIC and PIC-TD3 achieve a better balance of efficiency and performance.

## 4.3 GENERALIZATION

The advantage of PIC lies in its ability to leverage peer states and corresponding actions from the dataset, allowing the learning of potentially optimal actions not present in the current state. This gives PIC better generalization compared to traditional policy constraint methods. Inspired by (Ran et al., 2023) and (Kumar, Aviral , 2019), we create a lineworld environment. The environment details are in the Appendix E.2. We generate four datasets–*lineworld-easy*, *lineworld-random*, *lineworld-medium*, and *lineworld-hard*–ordered by increasing difficulty. We train TD3+BC (Fujimoto & Gu, 2021), BEAR (Kumar et al., 2019), PRDC (Ran et al., 2023), and PIC-TD3 on the datasets, and the results are presented in Appendix F.5.1. In easy and random datasets, all methods reach the optimal policy. On medium and hard datasets, reward sparsity and conservative regularization hinder TD3+BC and BEAR, while PRDC overlooks potential optimal actions. PIC generalizes by leveraging peer states' optimal actions, even unseen ones, to escape local optima.

Further, we systematically evaluate the impact of different state-space distance metrics on EPIC. We test multiple EPIC variants on both MuJoCo tasks and the high-dimensional quadruped benchmark Walk These Ways (WTW) Margolis & Agrawal (2023). The details of the WTW environment, its data collection procedure, and the complete results are provided in Appendix F.5.2. EPIC remains robust across all distance metrics in MuJoCo and continues to achieve competitive performance even in the high-dimensional WTW setting, demonstrating that its effectiveness stems from the peer-induced constraint mechanism rather than reliance on specific state-space metric.

## 4.4 PARAMETER STUDY

In this section, we conduct a parameter study on EPIC, focusing on the number of peer states $K$ (which affects the candidate action set), PIC strength $\delta$, and ensemble size $N$. We evaluate their sensitivity across various Gym-MuJoCo tasks, running for one million steps with five random seeds.

**Number of Peer States $K$:** The number of peer states $K$ is a key hyperparameter for PIC-TD3 and EPIC, influencing action diversity. As shown in Figure 4 (a) and (b), larger $K$ improves performance and accelerates convergence, but when $K > 20$ the gains plateau; PIC-TD3 slightly declines due to value estimation errors, while EPIC remains robust through ensemble regularization.

**PIC Strength $\delta$:** The PIC strength $\delta$ controls the intensity of the peer constraint. Larger $\delta$ enforces closeness to in-sample actions, limiting exploration, while smaller $\delta$ prioritizes Q-value fitting, risking deviation from the behavior policy. As shown in Figure 4 (c), performance on hopper-medium

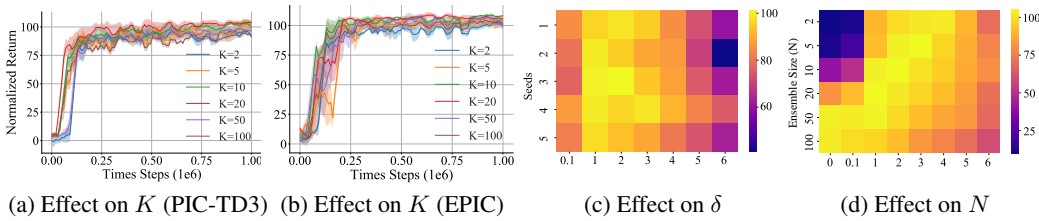

(a) Effect on $K$ (PIC-TD3)  (b) Effect on $K$ (EPIC)     (c) Effect on $\delta$           (d) Effect on $N$

Figure 4: The impact of peer states $K$, PIC strength $\delta$, and ensemble size $N$ on performance.

degrades when $\delta < 1$ or $\delta > 4$, with the best results in the moderate range $\delta \in [1, 3]$. Similar trends appear in AntMaze and Adroit (See Appendix F.6.1), indicating that proper tuning of $\delta$ ensures robust performance across tasks.

**Ensemble Size $N$:** Ensemble size $N$ improves OOD penalization but reduces efficiency. As shown in Figure 4 (d), performance on hopper-medium-replay improves with $N$ when $\delta = 0$, but declines for $N \geq 50$ with small $\delta$, and decreases further as $N$ grows under larger $\delta$. Similar trends are observed in additional experiments, reported in Appendix F.6.2.

These results highlight the Coupling Effect: moderate $\delta$ enhances generalization, but excessive $\delta$ amplifies uncertainty and weakens ensembles. A balanced choice ($\delta \in [1, 3]$, $N \in [5, 20]$) achieves robust performance.

### 4.5 Performance on Offline-to-Online D4RL

Evaluating offline-to-online performance is a critical aspect of offline RL algorithms, especially in light of recent advancements. Based on this, we conduct additional tests on EPIC, as it demonstrates remarkable effectiveness in offline pretraining through PIC and the Coupling Effect. As shown in Appendix F.7, EPIC achieves competitive results, outperforming in three of six AntMaze datasets and reaching state-of-the-art final scores on Adroit tasks.

## 5 Conclusion, Limitations, and Future Work

To address the limitations of existing offline RL methods in generalization and efficiency, we propose the Peer-Influenced Constraint (PIC) and its enhanced version, Ensemble Peer-Influenced Constraint (EPIC). PIC improves policy generalization by identifying potentially better actions from peer states. We also discover a direct relationship between PIC and uncertainty estimation, termed the Coupling Effect. Leveraging PIC and the Coupling Effect, EPIC achieves state-of-the-art performance with smaller ensemble sizes, reducing computational costs in offline RL. Extensive comparisons with strong baselines show that our method outperforms others across various tasks. We also validate its effectiveness during offline-to-online transitions.

We hope this work inspires further exploration of the interaction between constraint methods and uncertainty estimation, offering new insights to the community. A limitation of our approach is the inefficiency of peer state search in large-scale, high-dimensional datasets. We aim to develop more efficient offline RL methods for such datasets in future work.

ACKNOWLEDGMENTS

We would like to thank the anonymous reviewers for their very constructive comments. This work was supported by the National Natural Science Foundation of China (62276160), and the Natural Science Foundation of Shanxi Province, China (202203021211294).

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

## A  RELATED WORKS

Behavior Cloning (BC) (Bain & Sammut, 1995) is a common approach in policy-constrained offline RL. BCQ (Fujimoto et al., 2019) uses a Variational Autoencoder (Kingma & Welling, 2013) to model the behavior policy but is prone to fitting errors. TD3+BC (Fujimoto & Gu, 2021) integrates BC directly into policy loss, achieving competitive results without explicit behavior policy modeling. However, the distribution constraint of BCQ and TD3+BC often struggles to distinguish between optimal and suboptimal actions. BEAR (Kumar et al., 2019) and BRAC (Wu et al., 2019) constrain policy divergence via Kullback-Leibler or Wasserstein metrics. IQL (Kostrikov et al., 2021b) employs expectile regression and advantage-weighted BC without evaluating OOD actions. While these methods are efficient with limited networks, they often underperform ensemble-based approaches and are overly conservative. PRDC (Ran et al., 2023) enforces distance constraints using KD-Tree (Bentley, 1975) for neighbor search but faces challenges in balancing state-action priorities, manual parameter tuning, and large search spaces. Diffusion-DICE (Mao et al., 2024) transforms the behavior distribution into the optimal policy distribution using score decomposition and a guide-then-select strategy to avoid value exploitation errors. InAC (Xiao et al., 2023), SQL (Xu et al., 2023a), CDSA (Liu et al., 2024), and CPQ (Xu et al., 2022) incorporate in-sample constraints to avoid OOD actions, enhancing policy stability and generalization. These insights motivate our exploration of richer dataset utilization for precise behavior policy modeling and generalized policy learning.

In online RL, ensembles are used to build robust value estimates and enhance exploration (Wei et al., 2022; Zhang et al., 2024c). In offline RL, SAC-N (An et al., 2021) penalizes OOD actions via minimum ensemble values but requires up to 500 networks and 3M training steps. EDAC (An et al., 2021) reduces uncertainty and ensemble size by minimizing pairwise cosine similarity, promoting diversity among networks. RORL (Yang et al., 2022) smooths OOD action Q-values by learning uncertainties for different state-action pairs. MSG (Ghasemipour et al., 2022) strengthens pessimistic estimates by replacing shared targets with independent ones. PBRL (Bai et al., 2022) and MTDS (Bai et al., 2024) penalize value estimates based on action deviations from the mean. Though effective in uncertainty quantification, these methods often need large ensembles. SAC-RND (Nikulin et al., 2023) employs random network distillation with two Q-networks yet still incurs heavy training costs. Our method leverages the Coupling Effect to penalize OOD actions effectively with a smaller ensemble size, preserving data relationships and enabling generalized, efficient policy learning.

## B  THEORETICAL PROOFS

*Proof.* Since $P(s'|s,a)$ and $\mu$ are Lipschitz continuous, we have the following equations:

$$||P(s'|s,a_1) - P(s'|s,a_2)|| \leq L_P||a_1 - a_2||, \tag{13}$$

$$||\mu(s_1) - \mu(s_2)|| \leq L_s||s_1 - s_2||, \tag{14}$$

for all $s \in \mathcal{S}, a \in \mathcal{A}$. $L_P$ and $L_s$ represent the Lipschitz constants. The above equations have received significant attention in theoretical RL research (Ran et al., 2023; Saxena et al., 2023). Then, we have:

$$|J(\pi^*) - J(\pi)| = |J(\pi^*) - J(\mu) + J(\mu) - J(\pi)| \leq |J(\pi^*) - J(\mu)| + |J(\pi) - J(\mu)|. \tag{15}$$

Firstly, considering $|J(\pi) - J(\mu)|$, we define the occupancy measure:

$$d_{\pi_\phi}(s') = (1 - \gamma) \int_S \sum_{t=0}^{\infty} \gamma^t p_0(s) p(s \to s', t, \pi) \, \mathrm{d}s, \tag{16}$$

where $p(s \to s', t, \pi)$ represents the probability density of reaching state $s'$ from state $s$ after $t$ steps under policy $\pi$. Consequently, Equation (16) can be rewritten as:

$$J(\pi) = \frac{1}{1 - \gamma} \mathbb{E}_{s \sim d_{\pi_\phi}(s)}[r(s)]. \tag{17}$$

Then, we have

$$\begin{aligned}
|J(\pi) - J(\mu)| &= \left| \frac{1}{1-\gamma} \mathbb{E}_{s \sim d_{\pi_\phi}(s)}[r(s)] - \frac{1}{1-\gamma} \mathbb{E}_{s \sim d_\mu(s)}[r(s)] \right| \\
&= \frac{1}{1-\gamma} \left| \int_{\mathcal{S}} \left( d_{\pi_\phi}(s) - d_\mu(s) \right) r(s) \, ds \right| \\
&\leq \frac{1}{1-\gamma} \int_{\mathcal{S}} \left| d_{\pi_\phi}(s) - d_\mu(s) \right| |r(s)| \, ds \\
&\leq \frac{R_{\max}}{1-\gamma} \int_{\mathcal{S}} \left| d_{\pi_\phi}(s) - d_\mu(s) \right| \, ds \\
&\overset{(i)}{\leq} \frac{CL_P R_{\max}}{1-\gamma} \max_{s \in \mathcal{S}} \| \pi(s) - \mu(s) \| \\
&= \frac{CL_P R_{\max}}{1-\gamma} \max_{s \in \mathcal{S}} \| \pi(s) - \mu(\hat{s}_j) + \mu(\hat{s}_j) - \mu(s) \| \\
&\leq \frac{CL_P R_{\max}}{1-\gamma} \max_{s \in \mathcal{S}} \| \pi(s) - \mu(\hat{s}_j) \| + \max_{s \in \mathcal{S}} \| \mu(\hat{s}_j) - \mu(s) \| \\
&\leq \frac{CL_P R_{\max}}{1-\gamma} (\epsilon_a + L_s \epsilon_s).
\end{aligned} \tag{18}$$

Here, $(i)$ is due to $\int_{\mathcal{S}} \left| d_{\pi_\phi}(s) - d_\mu(s) \right| \, ds \leq CL_P \max_{s \in \mathcal{S}} \| \pi(s) - \mu(s) \|$, the proof of this inequality can be found in the appendix of (Xiong et al., 2022).

Then, considering $|J(\pi^*) - J(\mu)|$, we have

$$\begin{aligned}
|J(\pi^*) - J(\mu)| &= \left| \frac{1}{1-\gamma} \mathbb{E}_{s \sim d_{\pi_\phi^*}(s)}[r(s)] - \frac{1}{1-\gamma} \mathbb{E}_{s \sim d_\mu(s)}[r(s)] \right| \\
&= \frac{1}{1-\gamma} \left| \int_{\mathcal{S}} \left( d_{\pi_\phi^*}(s) - d_\mu(s) \right) r(s) \, ds \right| \\
&\leq \frac{1}{1-\gamma} \int_{\mathcal{S}} \left| d_{\pi_\phi^*}(s) - d_\mu(s) \right| |r(s)| \, ds \\
&\leq \frac{R_{\max}}{1-\gamma} \int_{\mathcal{S}} \left| d_{\pi_\phi^*}(s) - d_\mu(s) \right| \, ds \\
&\leq \frac{CL_P R_{\max}}{1-\gamma} \max_{s \in \mathcal{S}} \| \pi^*(s) - \mu(s) \| \\
&\leq \frac{CL_P R_{\max}}{1-\gamma} \epsilon_*.
\end{aligned} \tag{19}$$

Finally, combining Equation (18) and Equation (19), we have

$$\begin{aligned}
|J(\pi^*) - J(\pi)| &= |J(\pi^*) - J(\mu) + J(\mu) - J(\pi)| \\
&\leq |J(\pi^*) - J(\mu)| + |J(\pi) - J(\mu)| \\
&\leq \frac{CL_P R_{\max}}{1-\gamma} (\epsilon_a + L_s \epsilon_s + \epsilon_*).
\end{aligned} \tag{20}$$

$\square$

The proof of Theorem 1 is finished. From Theorem 1, the performance gap is influenced by $\epsilon_a$, $\epsilon_s$, and $\epsilon_*$. EPIC reduces $\epsilon_s$ by selecting peer states $\hat{s}_j$ that are close to the given state $s$, ensuring that the state space of the learned policy is well-aligned with the in-sample states. This minimizes the risk of state distribution shift. EPIC reduces $\epsilon_a$ by selecting actions $a^*$ that are close to both the behavior policy $\mu(s)$ and the optimal policy $\pi^*(s)$, using a combination of in-sample actions and the optimal policy. This ensures that the learned policy adheres to both the behavior and optimal policies, preventing large deviations in action selection. Finally, EPIC reduces $\epsilon_*$ by leveraging the structure of the dataset and using peer-influenced constraints to enforce the learned policy's alignment with the optimal policy. By effectively using peer states and action selections, EPIC narrows the performance gap and significantly enhances policy generalization, avoiding overfitting to the behavior policy and ensuring robust performance.

## C   PSEUDOCODE OF PIC-TD3

---

**Algorithm 2** TD3 Algorithm with Peer-Influenced Constraint (PIC-TD3)

---

**Input:** Observation state $\mathcal{S}$, offline dataset $\mathcal{D}$
**Output:** Learned policy $\pi$
Initialize: Critic networks $Q_{\theta_1}, Q_{\theta_2}$, actor network $\pi_\phi$, target networks $\theta_1' \leftarrow \theta_1, \theta_2' \leftarrow \theta_2, \phi' \leftarrow \phi$, and peer number $K$
**for** $t = 1$ to $T$ **do**
   Sample a mini-batch $\mathcal{B} = \{(s, a, r, s')\}$ from $\mathcal{D}$
   Find peer states in $\mathcal{D}$ of sample state by $\hat{s}_j = \arg\min_{\hat{s} \in \mathcal{D} \setminus \mathcal{D}_{j-1}} \|s - \hat{s}\|$
   Construct candidate action set $\mathcal{A}'$
   Find optimal action from $\mathcal{A}'$ by $a^* = \arg\max_{a \in \mathcal{A}'} \min_{i=1,2} Q_{\theta_i}(s, a)$
   Calculate PIC distance of sample state by $d_{\mathcal{D}}^{\text{PIC}}(s) = \|\pi_\phi(s) - a^*\|$
   Update $\theta_i$ by minimizing $L(\theta_i) = \mathbb{E}_{(s,a,r,s') \sim \mathcal{B}} \left[ (y - Q_{\theta_i}(s, a))^2 \right]$
   **if** $t \bmod d$ **then**
     Update $\phi$ using gradient descent with $L_{\text{PT}}(\phi) = \mathbb{E}_{s \sim \mathcal{B}} \left[ -\beta Q_{\theta_1}(s, a) \right] + \delta \mathbb{E}_{s \sim \mathcal{B}} \left[ d_{\mathcal{D}}^{\text{PIC}}(s) \right]$
     $\theta_i' \leftarrow \tau\theta_i + (1 - \tau)\theta_i'$, $\phi' \leftarrow \tau\phi + (1 - \tau)\phi'$
   **end if**
**end for**

---

## D   IMPLEMENTATION DETAILS

PIC-TD3 is implemented using PyTorch based on the TD3 implementation [1], and EPIC is implemented based on the author-provided implementation of EDAC [2].

The v2 version of D4RL benchmark datasets is utilized in Gym-MuJoCo and AntMaze tasks. For Adroit tasks, we use v1 version. The Gym-MuJoCo comparison results for CQL, SAC-N, and EDAC are sourced from (An et al., 2021), while the scores for other methods are taken from (Tarasov et al., 2023a). The AntMaze comparison results for CQL are taken from (Ghasemipour et al., 2022), while the scores for other methods are taken from (Tarasov et al., 2023a). The Adroit comparison results for BC, CQL, EDAC and RORL are taken from (Yang et al., 2022), while the scores for other methods are taken from (Tarasov et al., 2023a).

### D.1   SOFTWARE

We use the following hardware:

- Python 3.8
- D4RL 1.1 (Fu et al., 2020)
- MuJoCo 2.1.0 (Todorov et al., 2012)
- Gym 0.23.1 (Brockman et al., 2016)
- MuJoCo-py 2.1.2.14 (Todorov et al., 2012)
- PyTorch 2.1.2
- CUDA 11.8

### D.2   HARDWARE

- NVIDIA RTX 4090 $\times$ 4
- 14th Gen Intel(R) Core(TM) i9-14900K

---

[1] https://github.com/sfujim/TD3
[2] https://github.com/snu-mllab/EDAC

## D.3 HYPERPARAMETERS

The hyperparameters of the PIC and EPIC are detailed in Table 4 and Table 5.

Table 4: General hyperparameters of EPIC and PIC-TD3.

| Parameter | Value |
|---|---|
| **Shared** | |
| Optimizer | Adam |
| Learning rate | $3 \times 10^{-4}$ |
| Discount | 0.99 |
| Batch size | 256 |
| Size of hidden layers in critic and actor | 256 |
| Target network update rate $\tau$ | $5 \times 10^{-3}$ |
| Update-To-Data ratio | 1 |
| Nonlinearity | ReLU |
| $\alpha$ | 2.5 |
| **PIC-TD3** | |
| Target network update frequency $d$ | 2 |
| Action noise | $\mathcal{N}(0, 0.1)$ |
| Target policy smooth $\epsilon$ | $\text{clip}(\mathcal{N}(0, 0.2), -0.5, 0.5)$ |
| Critic number $M$ | 2 |

Table 5: Special EPIC hyperparameters on D4RL benchmark.

| Task Name | $N$ | $K$ | $\eta$ | $\delta$ | Task Name | $N$ | $K$ | $\eta$ | $\delta$ |
|---|---|---|---|---|---|---|---|---|---|
| halfcheetah-random-v2 | 2 | 10 | 1 | 1 | antmaze-umaze-v2 | 10 | 20 | 1 | 2 |
| halfcheetah-medium-v2 | 2 | 10 | 1 | 1 | antmaze-umaze-diverse-v2 | 10 | 20 | 1 | 2 |
| halfcheetah-expert-v2 | 2 | 10 | 1 | 2 | antmaze-medium-play-v2 | 10 | 20 | 1 | 2 |
| halfcheetah-medium-expert-v2 | 5 | 10 | 1 | 1 | antmaze-medium-diverse-v2 | 10 | 20 | 1 | 2 |
| halfcheetah-medium-replay-v2 | 5 | 10 | 1 | 1 | antmaze-large-play-v2 | 10 | 20 | 1 | 2 |
| halfcheetah-full-replay-v2 | 5 | 10 | 1 | 1 | antmaze-large-diverse-v2 | 10 | 20 | 1 | 2 |
| hopper-random-v2 | 10 | 10 | 1 | 1 | pen-human-v1 | 10 | 10 | 200 | 1 |
| hopper-medium-v2 | 10 | 10 | 1 | 1 | pen-cloned-v1 | 10 | 10 | 200 | 1 |
| hopper-expert-v2 | 10 | 10 | 1 | 2 | pen-expert-v1 | 10 | 10 | 200 | 1 |
| hopper-medium-expert-v2 | 10 | 10 | 1 | 2 | door-human-v1 | 10 | 10 | 200 | 3 |
| hopper-medium-replay-v2 | 10 | 10 | 1 | 1 | door-cloned-v1 | 10 | 10 | 200 | 3 |
| hopper-full-replay-v2 | 10 | 10 | 1 | 1 | door-expert-v1 | 10 | 10 | 200 | 3 |
| walker2d-random-v2 | 5 | 10 | 1 | 2 | hammer-human-v1 | 10 | 10 | 200 | 3 |
| walker2d-medium-v2 | 5 | 10 | 1 | 1 | hammer-cloned-v1 | 10 | 10 | 200 | 3 |
| walker2d-expert-v2 | 5 | 10 | 1 | 2 | hammer-expert-v1 | 10 | 10 | 200 | 3 |
| walker2d-medium-expert-v2 | 5 | 10 | 1 | 2 | relocate-human-v1 | 10 | 10 | 200 | 3 |
| walker2d-medium-replay-v2 | 5 | 10 | 1 | 2 | relocate-cloned-v1 | 10 | 10 | 200 | 3 |
| walker2d-full-replay-v2 | 5 | 10 | 1 | 1 | relocate-expert-v1 | 10 | 10 | 200 | 3 |

## E DETAILED ENVIRONMENT SETTING

### E.1 GRIDWORLD

In the gridworld environment, the agent's state is represented by its X-Y coordinates, and at each time step, it selects one of four possible actions: Up, Down, Left, or Right, with equal initial probabilities for each action. The agent only receives a reward of 1.0 when it successfully reaches the goal. Each episode terminates when the goal is reached or when the maximum episode length of 256 steps is reached. The layout of the grid world is shown in Figure 5.

The agent interacts with the environment for a total of $3 \times 10^5$ time steps, from which the last $2 \times 10^5$ transitions are used to form the offline dataset. Notably, we remove any experiences where the agent attempts to move right at the green triangle location. This exclusion is intentional, allowing us to test whether different methods can escape local optima and identify potentially better actions beyond those seen in the initial data.

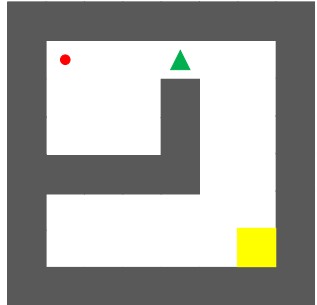

Figure 5: Visualization of gridworld.

### E.2 LINEWORLD

As shown in Figure 6, our lineworld environment consists of four task variations:

- lineworld-easy: In the dataset, the ratio of $-1$ to $+1$ actions for each state is $1 : 9$.

- lineworld-random: In the dataset, the ratio of $-1$ to $+1$ actions for each state is $1 : 1$.

- lineworld-medium: In the dataset, the ratio of $-1$ to $+1$ actions for each state is $9 : 1$.

- lineworld-hard: In the dataset, except for state 99, the ratio of $-1$ to $+1$ actions for each state is $99 : 1$. In state 99, only the action to move left is available.

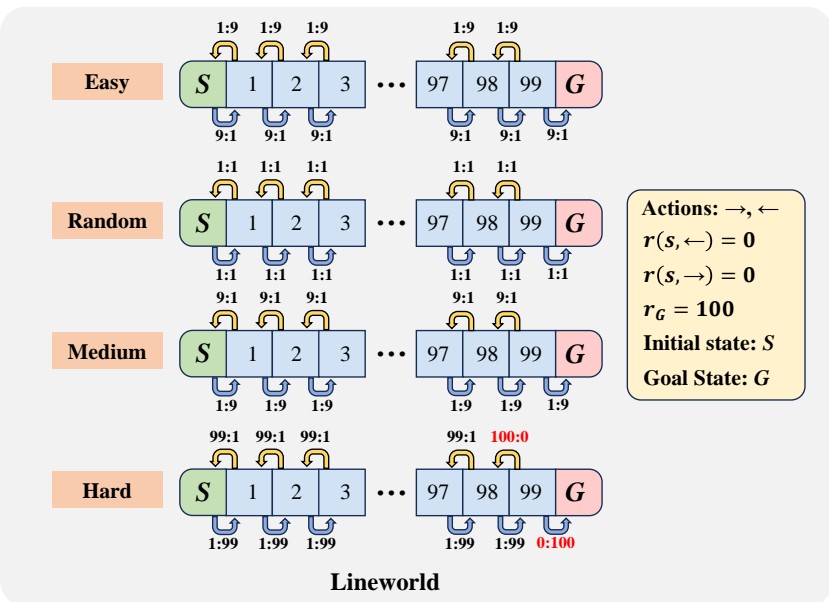

Figure 6: In the lineworld environment, the agent starts at position $S$ and aims to reach the goal at $G$. The action space consists of $[-1, +1]$, where $-1$ represents a move to the left, and $+1$ represents a move to the right. Reaching the goal ends the episode with a reward of $+100$, while all other actions result in a reward of $0$.

# F  MORE RESULTS

## F.1  COUPLING EFFECT

We measure and analyze the relationship between various uncertainty metrics, ensemble size $N$, and PIC strength $\delta$ on the hopper-medium-expert dataset. We sample 10,000 state-action pairs and compute the uncertainty metrics $Q_{\min}$, $Q_{\mathrm{std}}$, and $Q_{\mathrm{clip}} = Q_{\mathrm{mean}} - Q_{\min}$ at the same time step under different values of $N$ and $\delta$. We then calculate their average values and generate heatmaps, as shown in Figure 7. We find that, with a fixed $N$, increasing $\delta$ leads to higher uncertainty. Similarly, when $\delta$ is fixed, increasing $N$ also results in greater uncertainty. Considering that an excessively large $N$ can reduce computational efficiency, we can achieve effective uncertainty estimation with a moderate level of PIC strength.

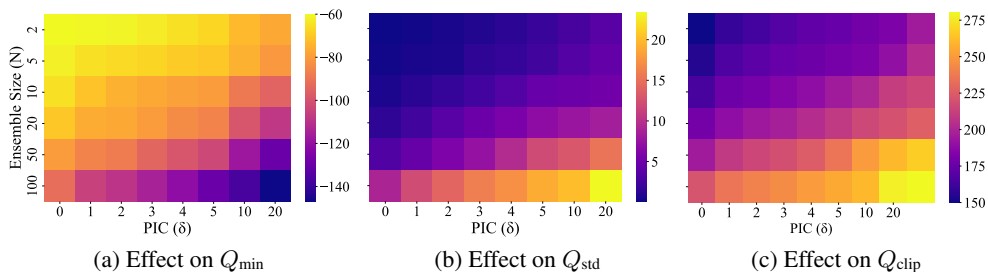

Figure 7: The impact of ensemble size $N$ and PIC strength $\delta$ on uncertainty metrics.

To further examine the influence of PIC strength on uncertainty modeling, we measure the evolution of $Q_{\min}$, $Q_{\mathrm{std}}$, and $Q_{\mathrm{clip}}$ across different training steps (10k–1M) and various values of $\delta$. The results are summarized in Figure 8. We observe that as $\delta$ and training progress increase, $Q_{\min}$ becomes increasingly pessimistic, while $Q_{\mathrm{std}}$ and $Q_{\mathrm{clip}}$ tend to rise on potential OOD action candidates, indicating stronger penalization on high-risk actions. These dynamics corroborate the proposed Coupling Effect: larger $\delta$ amplifies OOD penalties and yields more robust value estimation without requiring large ensemble sizes.

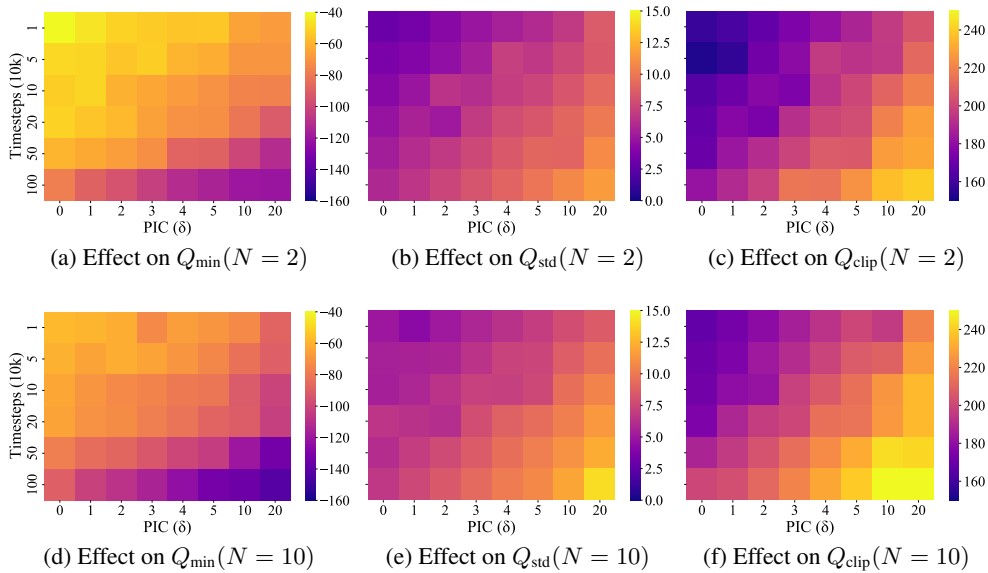

Figure 8: Uncertainty metric dynamics over training steps and PIC strengths $\delta$ for different ensemble sizes $N$.

In continuous action spaces, a policy action almost never matches a dataset action exactly. We therefore adopt a simple threshold-based criterion. We first estimate a global action-distance threshold $\tau_a$ from the offline dataset: for each dataset state, we retrieve its 5 nearest neighbor states, collect their behavior actions, and record the minimum action difference between the current action and these neighbor actions. The 90th percentile of these local minimum differences is taken as $\tau_a$, representing the typical scale of action variation in the dataset. During evaluation, we randomly sample 10,000 states from the policy's state distribution. For each sampled state, we directly compute the minimum distance between the policy action and all dataset actions. If this distance is below $\tau_a$, the action is classified as in-distribution; otherwise it is treated as OOD. The in-distribution ratio is defined as the fraction of sampled states satisfying this condition.

Using this criterion, we observe from Figure 9 that PIC significantly increases the in-distribution ratio compared to the no-PIC baseline, and this ratio continues to rise with larger PIC strength $\delta$, longer training horizons, and larger ensemble sizes $N$. This demonstrates that PIC progressively steers the learned policy back toward the dataset support, providing an additional perspective on the Coupling Effect discussed in the main text.

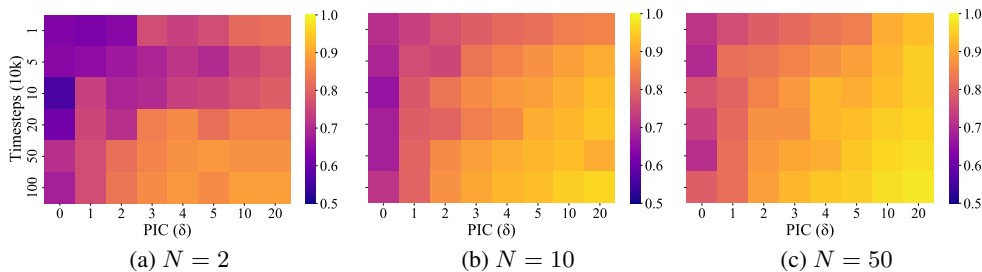

(a) $N = 2$        (b) $N = 10$        (c) $N = 50$

Figure 9: In-distribution action ratio over training steps and PIC strengths $\delta$ for different ensemble sizes $N$.

## F.2 MINIMUM NUMBER OF Q-ENSEMBLES

The minimum number of Q-ensembles for the results in Tables 1, 2, and 3 are in Figure 10. As $\delta$ increases, the required number of Q-ensembles decreases further. Notably, our method achieves competitive results with significantly fewer critics, demonstrating its efficiency and robustness. In Gym-MuJoCo tasks, m-e = medium-expert, m-r = medium-replay, and f-r = full-replay. In Adroit tasks, h = human, c = cloned, e = expert, ham = hammer, rel = relocate.

To further assess the robustness and fairness of our method, we additionally evaluate EPIC under a single shared hyperparameter configuration across all Gym-MuJoCo tasks. Specifically, we test two unified settings: EPIC-I with $N = 10$, $K = 10$, $\delta = 1$ and EPIC-II with $N = 10$, $K = 10$, $\delta = 2$. As shown in Table 6, EPIC under these unified hyperparameters still outperforms traditional baselines on most tasks, and its performance remains very close to the results reported in the main table, with only minimal differences. These results demonstrate that EPIC maintains stable and consistent performance even without task-specific adjustments, further supporting the robustness and practical usability of the proposed method.

## F.3 EXTENSION OF PIC TO IQL AND SAC

In addition to its application in TD3, we further extend the PIC framework to other RL algorithms, specifically IQL (Kostrikov et al., 2021b) and SAC (Haarnoja et al., 2018). This allows us to evaluate the versatility and scalability of PIC.

The results, shown in Table 7, demonstrate that PIC maintains its effectiveness in enhancing the generalization and robustness of the learned policies across a variety of environments. These findings highlight the scalability of the PIC framework, confirming its utility beyond TD3 and suggesting that it can be effectively applied to a wide range of RL algorithms.

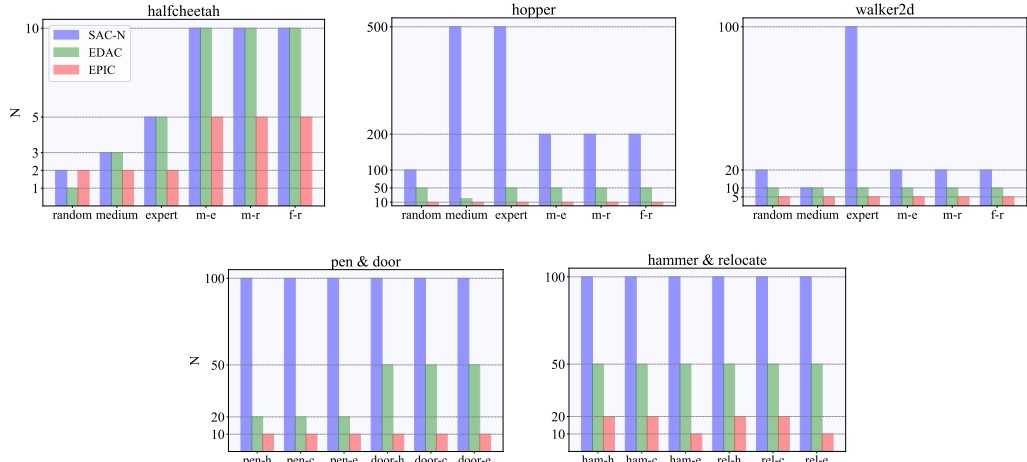

Figure 10: With $\delta = 1$, the minimum number of Q-ensembles required to attain comparable performance in Tables 1, 2, and 3.

Table 6: Average normalized score over final evaluations across five seeds on Gym-MuJoCo tasks.

| Task Name | SAC-RND | PRDC | SAC-N | EDAC | PIC-TD3 (ours) | EPIC (ours) | EPIC-I | EPIC-II |
|---|---|---|---|---|---|---|---|---|
| halfcheetah-random | 27.6 | 26.9 | 28.0 | 28.4 | $25.3 \pm 2.1$ | $\mathbf{28.9} \pm 1.4$ | $27.6 \pm 3.2$ | $25.0 \pm 4.1$ |
| halfcheetah-medium | 66.4 | 63.5 | 67.5 | 65.9 | $68.4 \pm 2.4$ | $\mathbf{68.9} \pm 2.7$ | $67.7 \pm 2.1$ | $65.2 \pm 2.4$ |
| halfcheetah-expert | 102.6 | – | 105.2 | 106.8 | $104.2 \pm 3.2$ | $107.9 \pm 6.4$ | $107.7 \pm 3.1$ | $\mathbf{108.5} \pm 4.8$ |
| halfcheetah-medium-expert | $\mathbf{108.1}$ | 94.5 | 107.1 | 106.3 | $99.8 \pm 1.6$ | $103.8 \pm 3.8$ | $102.4 \pm 3.2$ | $104.6 \pm 4.5$ |
| halfcheetah-medium-replay | 51.2 | 55.0 | 63.9 | 61.3 | $63.1 \pm 0.9$ | $\mathbf{64.8} \pm 1.5$ | $62.9 \pm 2.4$ | $63.3 \pm 2.1$ |
| halfcheetah-full-replay | 81.2 | – | 84.5 | 84.6 | $83.7 \pm 1.1$ | $87.3 \pm 1.0$ | $\mathbf{87.5} \pm 1.9$ | $85.5 \pm 2.3$ |
| hopper-random | 19.6 | 26.8 | $\mathbf{31.3}$ | 25.3 | $25.4 \pm 3.2$ | $27.9 \pm 4.2$ | $27.9 \pm 4.2$ | $28.3 \pm 3.1$ |
| hopper-medium | 91.1 | 100.3 | 100.3 | 101.6 | $100.7 \pm 1.5$ | $\mathbf{102.2} \pm 0.5$ | $\mathbf{102.2} \pm 0.5$ | $101.3 \pm 1.2$ |
| hopper-expert | 109.8 | – | 110.3 | 110.1 | $111.8 \pm 2.1$ | $\mathbf{112.2} \pm 1.5$ | $108.7 \pm 3.2$ | $\mathbf{112.2} \pm 1.5$ |
| hopper-medium-expert | 109.8 | 109.2 | 110.1 | 110.7 | $105.6 \pm 1.1$ | $112.3 \pm 0.9$ | $\mathbf{113.7} \pm 0.8$ | $112.3 \pm 0.9$ |
| hopper-medium-replay | 97.2 | 100.1 | 101.8 | 102.8 | $100.5 \pm 1.3$ | $102.0 \pm 1.0$ | $102.0 \pm 1.0$ | $\mathbf{103.5} \pm 1.8$ |
| hopper-full-replay | 107.4 | – | 102.9 | 105.4 | $105.8 \pm 0.8$ | $\mathbf{107.6} \pm 1.6$ | $\mathbf{107.6} \pm 1.6$ | $106.8 \pm 2.0$ |
| walker2d-random | 18.7 | 5.0 | 21.7 | 16.6 | $14.6 \pm 4.1$ | $22.4 \pm 1.6$ | $15.4 \pm 3.8$ | $\mathbf{23.7} \pm 2.1$ |
| walker2d-medium | 92.7 | 85.2 | 87.9 | 92.5 | $94.9 \pm 2.2$ | $95.9 \pm 3.2$ | $\mathbf{97.7} \pm 2.7$ | $96.7 \pm 4.3$ |
| walker2d-expert | 104.5 | – | 107.4 | 115.1 | $114.6 \pm 1.5$ | $\mathbf{117.7} \pm 2.1$ | $115.3 \pm 2.2$ | $116.4 \pm 0.9$ |
| walker2d-medium-expert | 104.6 | 111.2 | 116.7 | 114.7 | $114.4 \pm 0.9$ | $116.4 \pm 1.7$ | $\mathbf{117.3} \pm 1.4$ | $116.8 \pm 2.2$ |
| walker2d-medium-replay | 89.4 | 92.0 | 78.7 | 87.1 | $\mathbf{95.5} \pm 2.4$ | $95.2 \pm 2.2$ | $93.1 \pm 2.6$ | $\mathbf{95.5} \pm 3.3$ |
| walker2d-full-replay | 105.3 | – | 94.6 | 99.8 | $103.0 \pm 1.9$ | $\mathbf{107.4} \pm 2.2$ | $105.5 \pm 2.2$ | $105.4 \pm 1.8$ |
| **Average** | 82.6 | – | 84.4 | 85.2 | 85.1 | $\mathbf{87.8}$ | 86.8 | 87.3 |

## F.4 Performance and Efficiency Comparison

Time complexity plays a critical role in offline RL. We evaluate PIC-TD3, EPIC, and several baseline methods on the hopper-medium dataset. Figure 11 shows both the average training time (proportional to TD3+BC) and the corresponding normalized scores. Ensemble-based methods demonstrate strong performance but incur substantial computational overhead. In contrast, non-ensemble baselines such as TD3+BC, IQL, and CQL offer superior computational efficiency, though often at the cost of lower performance. EPIC successfully balances this trade-off, achieving both efficiency and robustness in policy learning. PIC-TD3, with its straightforward implementation and efficient search process, achieves performance comparable to ensemble methods while requiring significantly less training time.

## F.5 Generalization Results

### F.5.1 Lineworld

We generate four datasets–*lineworld-easy*, *lineworld-random*, *lineworld-medium*, and *lineworld-hard*–ordered by increasing difficulty. In *lineworld-easy*, the ratio of $-1$ to $+1$ actions is $1 : 9$. In *lineworld-random*, the ratio is balanced at $1 : 1$. In *lineworld-medium*, the ratio skews to $9 : 1$ for $-1$ actions. Finally, *lineworld-hard* has a $99 : 1$ ratio of $-1$ to $+1$ actions, except for state 99, where

Table 7: Average normalized score over the final evaluation and five seeds on the AntMaze tasks.

| Task Name | CQL | SAC-RND | PRDC | MSG | SAC-BC-N | PIC-SAC (ours) | PIC-IQL (ours) | PIC-TD3 (ours) | EPIC (ours) |
|---|---|---|---|---|---|---|---|---|---|
| antmaze-umaze | 74.0 | 97.0 | **98.8** | 97.9 | 98.6 | 95.4±2.2 | 97.9±1.5 | 96.4 ± 3.5 | 98.6 ± 0.5 |
| antmaze-umaze-diverse | 84.0 | 66.0 | 90.0 | 79.3 | 91.2 | 93.2±2.1 | 87.6±3.6 | 91.1 ± 2.3 | **94.3** ± 4.2 |
| antmaze-medium-play | 61.2 | 38.5 | 82.8 | 85.9 | 85.8 | 82.4±3.7 | 84.5±2.8 | 79.3 ± 5.4 | **88.1** ± 3.3 |
| antmaze-medium-diverse | 53.7 | 74.7 | 78.8 | **84.6** | 73.8 | 75.5±4.9 | 80.2±3.3 | 77.9 ± 6.8 | 83.8 ± 4.0 |
| antmaze-large-play | 15.8 | 43.9 | 54.8 | 64.3 | 65.8 | 62.3±3.9 | 62.3±2.1 | 55.7 ± 7.7 | **65.9** ± 2.3 |
| antmaze-large-diverse | 14.9 | 45.7 | 50.0 | 71.3 | **75.8** | 47.6±4.2 | 59.6±3.1 | 53.2 ± 4.7 | 66.9 ± 5.3 |
| **Average** | 50.6 | 61.0 | 75.9 | 80.6 | 81.8 | 76.1 | 78.7 | 75.6 | **82.9** |

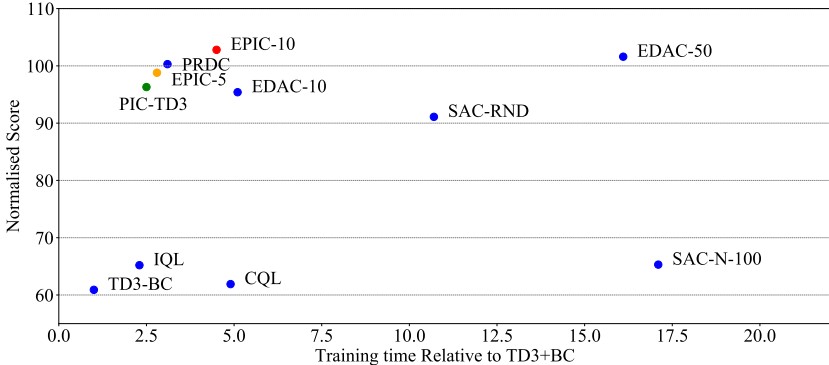

Figure 11: Comparison of performance and efficiency across different methods on hopper-medium. Here, method names such as EPIC-5, EPIC-10, EDAC-10, and EDAC-50 indicate the number of critics used in each ensemble.

only the −1 action is available. These datasets vary in difficulty based on the action distributions, offering a diverse set of challenges for the agent.

We train TD3+BC, BEAR, PRDC, and PIC-TD3 on the datasets, as shown in Figure 12. In easy and random datasets, all methods learn the optimal policy. However, with medium and hard datasets, reward sparsity and conservative policy regularization prevent TD3+BC and BEAR from achieving the optimal policy. While PRDC leverages dataset state-action pairs, it ignores potential optimal actions. PIC uniquely generalizes by exploiting optimal actions of peer states, even those unseen in current states, escaping local optima.

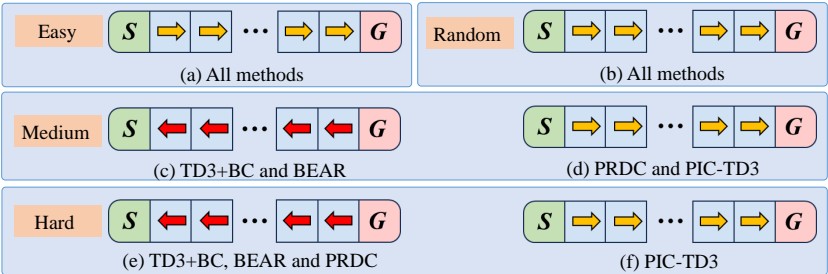

Figure 12: Visualization of the learned policy of different methods on four lineworld datasets.

### F.5.2 ANALYSIS OF STATE-SPACE DISTANCE METRICS

To comprehensively examine the influence of state-space distance metrics and representation methods on EPIC, we conduct extensive experiments on six MuJoCo tasks (hopper-medium, hopper-expert, hopper-medium-expert, walker2d-medium, walker2d-expert, walker2d-medium-expert) and on the high-dimensional quadruped locomotion benchmark Walk These Ways (WTW) (Margolis & Agrawal, 2023).

WTW is a realistic, high-dimensional, and strongly multimodal quadruped control platform built upon the Multiplicity of Behavior framework. Following the reference implementation,[3] we set the state dimension to 70 to balance realism and computational tractability. WTW provides a behavior-conditioned interface that enables a single controller to generate diverse locomotion behaviors, including pacing, trotting, bounding, pronking, and crouching.

We construct a velocity-tracking task using three representative gaits: pacing, trotting, and bounding. For each gait, we collect 4096 episodes of 250 time steps each, forming a large-scale, high-dimensional, and multimodal offline dataset Yuan et al. (2024). Performance is evaluated using the following metrics:

- **Stability**: fraction of episodes without falling within 250 steps;
- **Average x-axis speed**: average forward velocity.

To analyze the effect of distance metrics and state representations on nearest-neighbor retrieval, we evaluate several EPIC variants:

- **CQL** (Kumar et al., 2020) and **EDAC** (An et al., 2021) as strong value-regularization baselines;
- **EPIC-Raw**: KD-tree built on raw states using Euclidean distance;
- **EPIC-Norm**: per-dimension min–max normalization before KD-tree construction;
- **EPIC-PCA**: PCA applied to standardized states, retaining principal components that explain at least 99% of the cumulative variance;
- **EPIC-Embed**: a two-layer MLP autoencoder trained to map states into a 64-dimensional latent space, where the KD-tree is built (all other EPIC components remain unchanged).

As shown in Tables 8 and 9, EPIC achieves highly consistent performance across Raw, Norm, PCA, and Embed variants on MuJoCo tasks. In these medium-dimensional and well-structured state spaces, PCA and normalization introduce only mild transformations, and the autoencoder typically learns an almost linear embedding. As a result, the neighborhood structure is largely preserved, making EPIC insensitive to the particular choice of distance metric.

In the high-dimensional and multimodal WTW environment, EPIC's overall performance decreases because raw Euclidean distance can be dominated by a few high-variance dimensions, reducing neighbor quality. Nonetheless, EPIC still outperforms CQL and remains comparable to EDAC (3M steps). Among the variants, EPIC-Norm provides limited improvements through scale alignment; EPIC-PCA produces more reliable neighborhoods by capturing dominant variation directions; and EPIC-Embed performs best, as its learned latent space clusters states with similar gait and forward-velocity characteristics.

Overall, the benefits of EPIC arise primarily from the peer-induced constraint mechanism rather than dependence on a specific metric. While raw Euclidean distance is already effective on standard MuJoCo benchmarks, introducing mild representation learning can further improve neighbor quality and policy performance in high-dimensional or strongly multimodal environments.

Table 8: Effect of state similarity metrics on EPIC across MuJoCo tasks (five seeds).

| Task Name | CQL | EDAC | EPIC-Raw | EPIC-Norm | EPIC-PCA | EPIC-Embed |
|---|---|---|---|---|---|---|
| hopper-medium | 61.9 | 101.6 | 102.2 ± 0.5 | 101.4 ± 0.8 | 101.5 ± 2.7 | **103.1** ± 1.6 |
| hopper-expert | 106.5 | 110.1 | 112.2 ± 1.5 | 113.1 ± 1.2 | **113.4** ± 1.3 | 111.0 ± 1.4 |
| hopper-medium-expert | 96.9 | 110.7 | 112.3 ± 0.9 | **112.5** ± 3.1 | 111.8 ± 1.0 | 108.5 ± 2.8 |
| walker2d-medium | 79.5 | 92.5 | 95.9 ± 3.2 | 96.5 ± 2.7 | **96.8** ± 2.0 | 93.2 ± 1.9 |
| walker2d-expert | 109.3 | 115.1 | **117.7** ± 2.1 | 116.9 ± 3.1 | 114.9 ± 2.9 | 116.3 ± 1.8 |
| walker2d-medium-expert | 109.1 | 114.7 | 116.4 ± 1.7 | **117.7** ± 1.4 | 115.1 ± 1.6 | 117.2 ± 0.9 |
| **Average** | 93.9 | 107.5 | 109.5 | **109.7** | 108.9 | 108.2 |

---

[3]https://github.com/Improbable-AI/walk-these-ways

Table 9: Average speed (m/s) and stability (%) comparison of different state-similarity metrics for EPIC in quadruped locomotion velocity-tracking (WTW) tasks (three seeds).

| | CQL | EDAC | EPIC-Raw | EPIC-Norm | EPIC-PCA | EPIC-Embed |
|---|---|---|---|---|---|---|
| | Speed / Stability | Speed / Stability | Speed / Stability | Speed / Stability | Speed / Stability | Speed / Stability |
| Pacing (0.5 m/s) | 0.31 / 37.21 | 0.38 / 41.52 | 0.33 / 40.28 | **0.41** / 64.35 | 0.36 / 61.62 | 0.37 / **67.35** |
| Trotting (0.5 m/s) | 0.33 / 35.53 | 0.30 / 39.41 | 0.36 / 37.44 | 0.32 / 50.37 | 0.35 / 57.35 | **0.39** / 58.97 |
| Bounding (0.5 m/s) | 0.24 / 21.66 | 0.36 / 47.65 | 0.32 / 30.57 | 0.44 / 50.12 | 0.39 / **53.67** | **0.45** / 52.45 |
| Pacing (1.0 m/s) | 0.39 / 44.64 | 0.44 / 51.67 | 0.53 / 54.68 | 0.52 / 61.45 | 0.55 / **71.61** | **0.56** / 71.57 |
| Trotting (1.0 m/s) | 0.33 / 47.37 | 0.31 / 40.34 | 0.41 / 50.76 | **0.47** / 46.66 | 0.43 / 54.63 | 0.46 / **55.54** |
| Bounding (1.0 m/s) | 0.28 / 32.45 | 0.51 / 45.52 | 0.40 / 34.68 | 0.42 / 43.69 | 0.47 / 63.48 | **0.55** / **65.79** |

## F.6 PARAMETER STUDY

### F.6.1 PIC STRENGTH $\delta$

The PIC strength $\delta$ directly controls the intensity of the peer constraint. A larger $\delta$ places more emphasis on keeping the learned policy close to the in-sample optimal action, potentially reducing exploration. Conversely, a smaller $\delta$ focuses more on fitting Q-values, which may lead to greater deviations from the behavior policy. As shown in Figure 13, in the Gym-MuJoCo environments the algorithm's performance degrades markedly when the peer-influenced constraint strength $\delta$ is set too low ($< 1$) or too high ($> 4$). In contrast, the best results are consistently observed within a moderate range ($\delta \in [1, 3]$). A similar trend holds for the AntMaze and Adroit benchmarks, indicating that appropriately tuning $\delta$ yields robust performance across diverse task types.

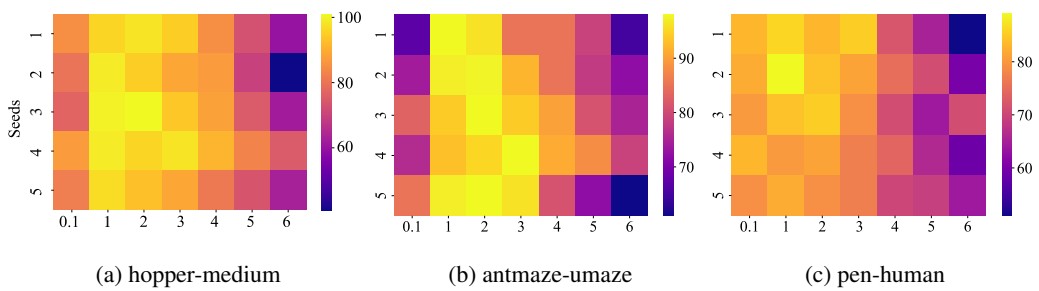

(a) hopper-medium        (b) antmaze-umaze        (c) pen-human

Figure 13: Impact of PIC strength $\delta$ on PIC-TD3 performance.

### F.6.2 ENSEMBLE SIZE $N$

Ensemble size $N$ controls uncertainty estimation strength. Larger $N$ helps penalize OOD actions in ensemble methods, but too large a $N$ reduces efficiency. As shown in Figure 14, without PIC ($\delta = 0$), performance improves with $N$, but for small $\delta$, performance starts declining when $N \geq 50$. For larger $\delta$, performance decreases as $N$ increases.

To further assess the sensitivity of EPIC to different action-selection rules, we conduct an additional study on hopper-medium, comparing the standard min-over-critics, Random Double Q (sampling two critics from $N = 10$), and Mean-Q aggregation under varying ensemble sizes $N \in \{1, 2, 10, 50\}$. The results (Figure 15) show that once $N \geq 2$, all three variants exhibit similar performance with only minor deviations, suggesting that EPIC is largely insensitive to the specific critic aggregation rule. When $N$ becomes very large (e.g., $N = 50$), learning slows down due to overly pessimistic value estimates, consistent across all variants. When $N = 1$, the performance degrades to some extent. However, under the peer-influenced constraint, the policy remains supported within the dataset distribution and can still learn a moderately good policy. In contrast, other single-critic networks typically fail in this setting Wei et al. (2022).

## F.7 PERFORMANCE ON OFFLINE-TO-ONLINE D4RL

Evaluating offline-to-online performance is a critical aspect of offline RL algorithms, especially in light of recent advancements. Based on this, we conduct additional tests on EPIC, as it demonstrates

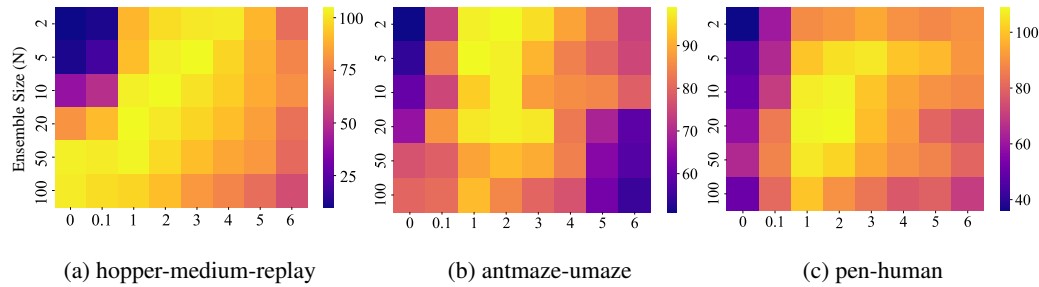

(a) hopper-medium-replay      (b) antmaze-umaze      (c) pen-human

Figure 14: Impact of ensemble size $N$ on EPIC performance.

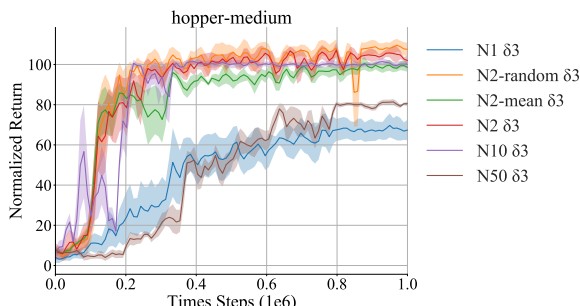

Figure 15: Performance of different action-selection strategies under varying ensemble sizes $N$.

remarkable effectiveness in offline pretraining through PIC and the Coupling Effect. We follow the methodology of (Tarasov et al., 2023b) to assess EPIC's performance in offline-to-online transitions. We compare EPIC's performance with TD3+BC (Fujimoto & Gu, 2021), IQL (Kostrikov et al., 2021b), and SPOT (Wu et al., 2022) during the transition from offline to online learning. Baselines scores are taken form (Tarasov et al., 2023a).

The results, including scores from the offline phase and after online fine-tuning, are presented in Table 10. EPIC demonstrates competitive performance, outperforming in three out of six AntMaze datasets and achieving state-of-the-art results in the final scores for the Adroit tasks.

Table 10: Normalized performance after offline pretraining and online fine-tuning on D4RL.

| Task Name | TD3 + BC | IQL | SPOT | EPIC (ours) |
|---|---|---|---|---|
| antmaze-umaze | 66.8 → 91.4 | 77.0 → 96.5 | 91.0 → 99.5 | **98.6 → 99.5** |
| antmaze-umaze-diverse | 59.1 → 48.4 | 59.5 → 63.8 | 36.3 → 95.0 | **94.3 → 98.3** |
| antmaze-medium-play | 59.2 → 49.4 | 71.8 → 89.8 | 71.8 → 97.5 | **88.1 → 95.4** |
| antmaze-medium-diverse | 62.6 → 49.4 | 64.3 → 92.3 | 73.8 → **94.5** | **83.8** → 93.1 |
| antmaze-large-play | 21.5 → 0.1 | 38.5 → 64.3 | 31.8 → **97.3** | **65.9** → 89.3 |
| antmaze-large-diverse | 9.5 → 0.4 | 26.8 → 64.3 | 17.5 → **81.0** | **66.9** → 74.8 |
| **AntMaze Average** | 46.5 → 40.0 (-6.5) | 56.3 → 78.5 (+22.2) | 53.7 → **94.1** (+40.4) | **82.9** → 91.7 (+8.8) |
| pen-cloned | 86.1 → 110.3 | 84.2 → 102.0 | 6.2 → 43.6 | **94.6 → 125.4** |
| door-cloned | 0.0 → 3.4 | 1.2 → 20.3 | 2.0 → 5.7 | **7.1 → 21.2** |
| hammer-cloned | 2.4 → 11.6 | 1.4 → 57.3 | 4.0 → 3.7 | **12.7 → 64.3** |
| relocate-cloned | -0.1 → 0.1 | -0.2 → -0.2 | -0.2 → -0.2 | **1.1 → 2.0** |
| **Adroit Average** | 22.1 → 31.4 (+9.3) | 21.7 → 44.9 (+23.2) | 3.0 → 13.2 (+10.2) | **28.9 → 53.2** (+24.4) |
| **Total Average** | 36.7 → 36.5 (-0.2) | 42.5 → 65.0 (+22.5) | 33.4 → 61.8 (+28.4) | **63.4 → 77.7** (+14.3) |

## F.8 TRAINING CURVES

We present the training curves of the algorithms in our experiments. The solid lines in the figure represent the average performance across five random seeds, while the shaded areas indicate the standard deviation, providing a measure of variability in the results.

In Figure 16, we display the training curves of EPIC and EDAC (An et al., 2021) over three million steps across 18 tasks in MuJoCo. EPIC is implemented based on the author-provided implementation of EDAC [4]. In Figure 17, we further compare the Interquartile Mean (IQM) values of EPIC and EDAC. IQM is a more reliable metric than simple averages, as it takes the distribution of outcomes into account and better reflects the overall performance, especially in terms of risk and outlier handling. Additionally, we plot the training curves for the AntMaze and Adroit environments, where we compare EPIC against IQL (Kostrikov et al., 2021b), a stable baseline known for its robust performance, as shown in Figures 18 and 19. IQL is implemented based on the CORL codebase [5].

---

[4]https://github.com/snu-mllab/EDAC
[5]https://github.com/tinkoff-ai/CORL

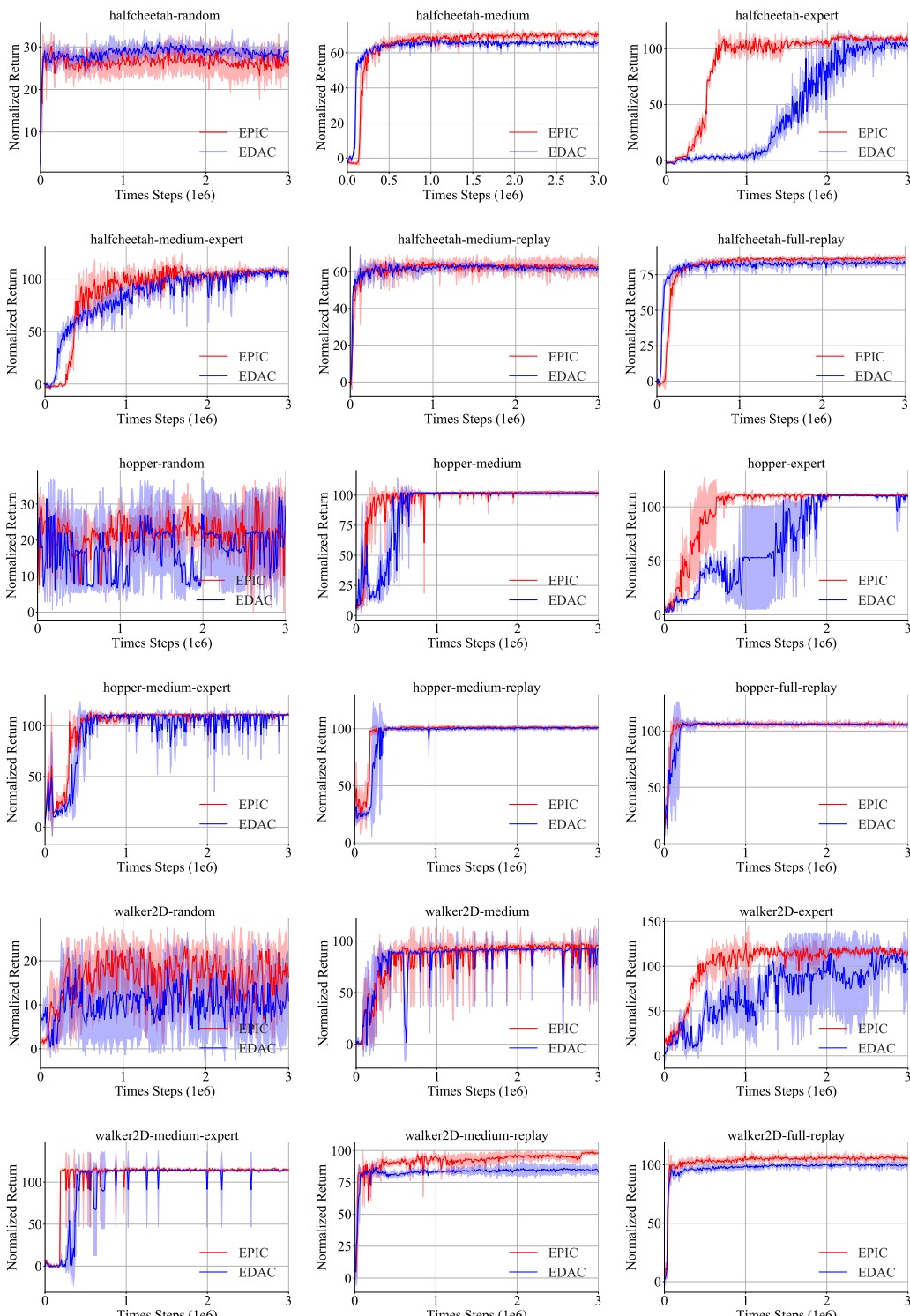

Figure 16: Comparative mean performance analysis of EPIC and EDAC across 18 MuJoCo tasks.

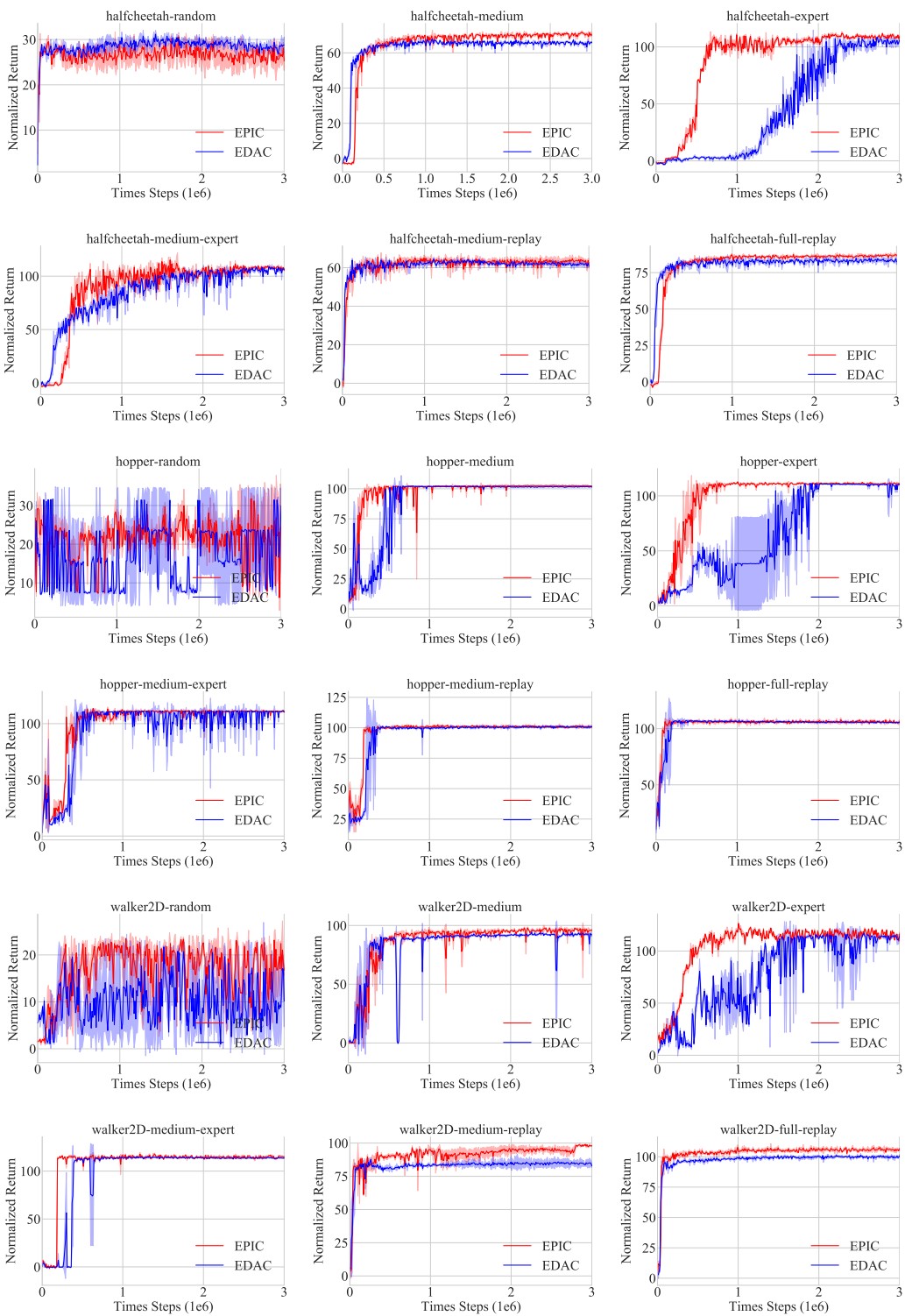

Figure 17: Comparative IQM performance analysis of EPIC and EDAC across 18 MuJoCo tasks.

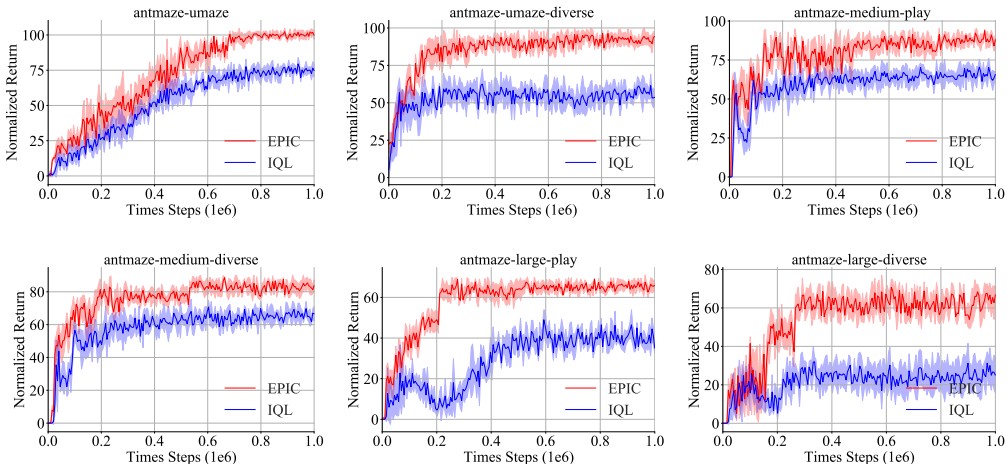

Figure 18: Comparative mean performance of EPIC and IQL across six AntMaze tasks.

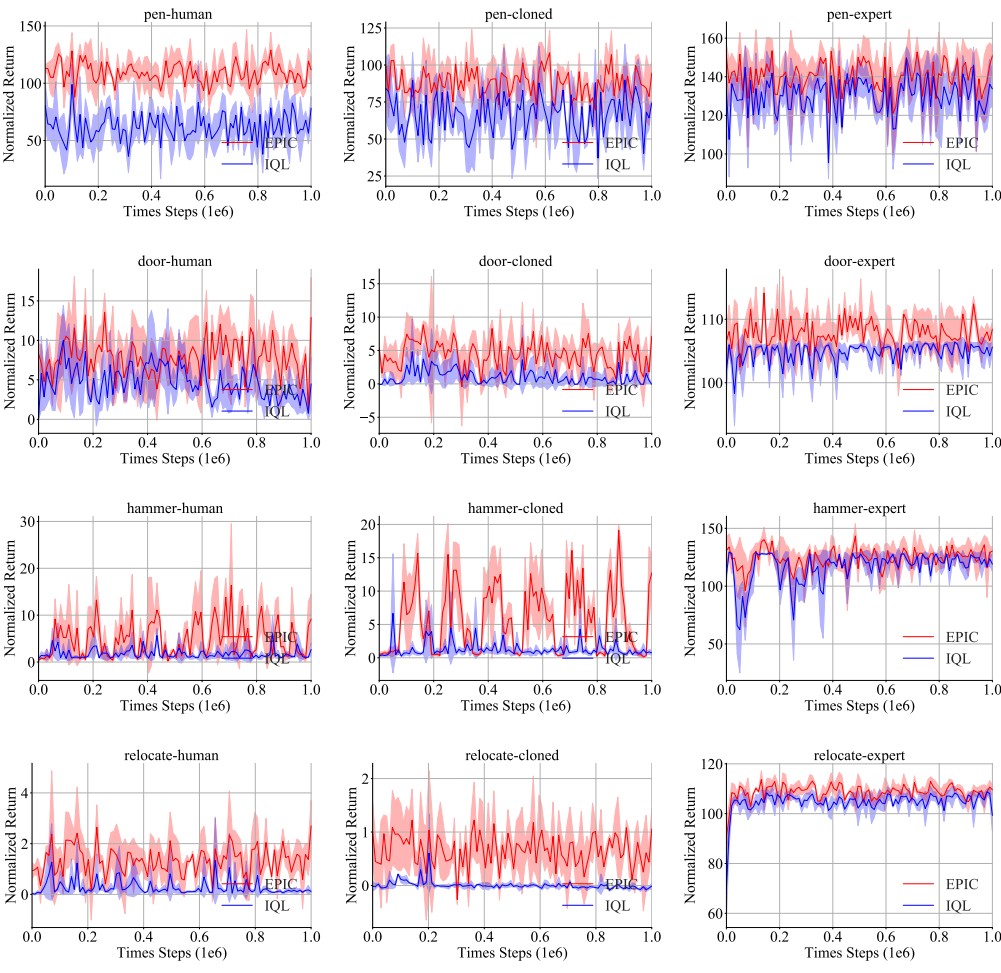

Figure 19: Comparative mean performance of EPIC and IQL across 12 Adroit tasks.

## G   LIMITATIONS AND FUTURE WORK

Despite promising results, several limitations remain. (i) Peer selection currently relies on Euclidean nearest neighbors in the raw state space; in high-dimensional, highly nonlinear, or visual settings, Euclidean similarity may diverge from semantic similarity, yielding "pseudo" peers. (ii) Candidate actions are drawn from local neighborhoods of the dataset and thus are sensitive to data sparsity or distributional skew, which can expose coverage gaps. (iii) The best-performing hyperparameters (e.g., PIC strength $\delta$) vary across tasks, making selection nontrivial and potentially brittle.

Future work includes: (i) retrieving peers in a learned representation space via contrastive or metric learning (optionally action-conditioned), replacing raw Euclidean distance; and (ii) developing adaptive schemes for automatically tuning $\delta$ (e.g., uncertainty- or constraint-driven adjustment) to improve robustness across diverse environments.

## H   USE OF LARGE LANGUAGE MODELS

Large Language Models were only used for minor language polishing. They were not involved in method design, theoretical analysis, or experiments, and therefore do not affect the originality of this work.

