# OpenReview forum: "Efficient Offline Reinforcement Learning via Peer-Influenced Constraint"
_ICLR.cc/2026/Conference — ICLR 2026 Poster_

### Official Review · Reviewer_dFV9 · 2025-11-01

**Soundness:** 3
**Presentation:** 3
**Contribution:** 3
**Rating:** 4
**Confidence:** 4

**Summary:**

This paper proposes a new regularizer for policy optimization in offline RL called PIC. Unlike conventional approaches that constrain the policy toward the specific action observed for a given state, PIC encourages the policy to align with actions drawn from a set of similar (peer) states, which can be interpreted as a relaxation of the conventional approaches. The experiment results show the proposed method achieves state-of-the-art results on D4RL benchmarks.

**Strengths:**

- The proposed method is intuitive and tackles a valid limitation of the commonly used policy regularizer in offline RL.
- The paper provided a theoretical justification of the proposed approach.
- The paper achieves strong empirical results across multiple tasks in the D4RL benchmark.
- The proposed method can be integrated into other (offline) RL methods like TD3 or IQL.

**Weaknesses:**

- The peer identification relies on norm distance in the raw state space, which may not align with semantic similarity in high-dimensional (for example, pixel) environments. The proposed method should be tested in such environments.
- Figure 4 (c) seems to show that the proposed method is somewhat sensitive to the hyperparameter delta.
- Table 5 shows the proposed method used different hyperparameters for each task in D4RL Mujoco. However, as far as I know, the baselines like CQL and IQL used the same hyperparameters for the Mujoco tasks. Therefore, this is not a fair comparison.

**Questions:**

- In Equation (7), it seems when j=1, \hat{s}_j will always be s. Is this correct?
- In Equation (9), why are alpha and delta both required? To my understanding, only one of the two is required to adjust the importance between the two terms.

---

> ### Author Response · Authors · 2025-11-22
> **Response to Reviewer dFV9 (1/2)**
>
> We sincerely thank the reviewer for the careful evaluation and constructive feedback, and we address each weakness and question point-by-point below.
>
> ---
>
> ### **Regarding W1: Using Norm Distance in Raw State Space for Peer Identification**
>
> Thank you for the insightful comment. We would like to clarify that **PIC does not rely on any special property of Euclidean distance**, but instead on the broader principle that “locally similar states provide useful candidate actions,” making the method fundamentally **metric-agnostic**. To directly evaluate this, **Appendix F.5.2** includes a comprehensive comparison of four state representations:
>
> - Raw Euclidean distance
> - Per-dimension min–max normalization
> - PCA (≥99% variance retained)
> - A 64-dimensional learned autoencoder embedding
>
> Across six MuJoCo tasks and the high-dimensional, multimodal WTW benchmark, we observe:
>
> - On **MuJoCo**, all four variants behave almost identically, indicating that EPIC is **highly robust to the choice of metric**;
> - On **WTW**, PCA and learned embeddings yield moderate improvements, demonstrating that PIC can **naturally integrate with representation learning to enhance neighbor quality** without modifying the algorithm.
>
> These findings show that the **effectiveness of EPIC arises from the peer-influenced constraint rather than reliance on raw Euclidean geometry**. While raw pixels do introduce semantic distortion, PIC can seamlessly operate on learned embeddings (CNN/RL encoders, self-supervised features, etc.), making it directly extendable to pixel-based settings.
>
> ---
>
> ### **Regarding W2: Sensitivity to the Hyperparameter $\delta$ in Figure 4 (c)**
> Thank you for raising this point. **Figure 4 (c)** and **Appendix F.6.1** indeed show that performance **decreases somewhat** when $\delta$ is too small (<1) or too large (>4). We offer the following clarifications.
>
> **1. A broad and stable effective region.**
> Across Gym-MuJoCo, AntMaze, and Adroit, **the best or near-best performance consistently occurs within **$\delta \in [1,3]$**, with strong agreement across seeds**. Moreover, Figure 14 shows that when the ensemble size is small, the Coupling Effect between PIC strength and uncertainty makes $\delta$ even less sensitive—yielding smoother $\delta$–performance curves compared to large-$N$ settings.
>
> Together, these results indicate that PIC exhibits a **wide and stable region of robust behavior**, rather than requiring fine-grained parameter tuning.
>
> **2. The degradation at extremes is expected and interpretable.**
> Small $\delta$ weakens the peer constraint, whereas large $\delta$ overly restricts exploration. The observed results is consistent with the intended role of $\delta$ and reflects a predictable regularization trade-off, rather than irregular or unstable sensitivity.
>
> **3. Visual contrast is amplified by the heatmap scale.**
> The heatmap uses a **40–100** range rather than 0–100, which makes color differences appear sharper than the underlying numerical variation.
>
> Overall, while $\delta$ naturally influences the balance between regularization and value fitting, the method exhibits a **wide stable region and consistent trends across tasks**. We therefore view its tuning difficulty as comparable to that of standard regularization coefficients commonly used in offline RL.
>
> ---
>
> ### **Regarding Weakness 3: Fairness of Comparison**
>
> Thank you for the comment. The hyperparameters reported in Table 5 were **not chosen via per-task tuning**, but rather to balance computational cost and performance (e.g., larger $N$ significantly increases computation). This led to small configuration differences across tasks.
>
> To directly address fairness, we re-evaluated EPIC on all MuJoCo tasks using **exactly the same hyperparameters**:
>
> - **EPIC-I:** $N = 10$, $K = 10$, $\delta = 1$
> - **EPIC-II:** $N = 10$, $K = 10$, $\delta = 2$
>
> As shown in **Appendix F.2**:
>
> - EPIC with shared hyperparameters still outperforms strong baselines on most tasks;
> - Its performance is very close to the results in Table 1, despite removing all configuration differences;
> - This suggests that while hyperparameters naturally have some effect, the **overall improvements are largely driven by the advantages of our peer-influenced constraint, which provides a stable and effective form of guidance across different tasks**.
>
> Therefore, we believe the comparison is fair, and EPIC demonstrates strong robustness even under fully unified settings.

---

> > ### Author Response · Authors · 2025-11-22
> > **Response to Reviewer dFV9 (2/2)**
> >
> > ### **Regarding Q1: Does Eq. (7) Incorrectly Select the Current State When $j=1$?**
> >
> > Thank you for pointing this out. Under the original notation, if $\mathcal{D}_{0}=\emptyset$, then for $j=1$ the minimization in Eq. (7) is indeed taken over the full dataset, and the current state $s$ could be selected because $\||s - \hat{s}\|| = 0$. We agree that the notation can be made more precise.
> >
> > In our implementation, however, **the current state $s$ is always explicitly excluded from peer retrieval**, since its own actions are already included separately in the candidate set. The precise formulation is:
> >
> > $\hat{s}_ j = \arg\min_{\hat{s} \in \mathcal{D} \setminus (\mathcal{D}_{j-1} \cup \lbrace s \rbrace)} ||s - \hat{s}||.$
> >
> > We have updated the manuscript to use this corrected formulation so that the notation fully matches the implementation. **Importantly, this clarification does not affect any empirical results, as the implementation always uses $K$ peer states distinct from $s$.**
> >
> > ---
> >
> >
> > ### **Regarding Q2: Why are Both $\alpha$ (Inside $\beta$) and $\delta$ Needed?**
> >
> > Thank you for the question. Although both appear as weights in the actor objective, they regulate **orthogonal aspects** of the update and are not interchangeable.
> >
> > **1. $\alpha$ (through $\beta$) is inherited from TD3+BC and normalizes the numerical scale of Q-values.**
> > Q-values can vary widely across tasks and training stages; without normalization, the value term can either dominate or vanish. TD3+BC therefore uses $\beta$ to stabilize training. Since PIC-TD3 is built directly on TD3+BC, we **keep this mechanism unchanged** to preserve its numerical stability and ensure a fair comparison. This Q-scale normalization is orthogonal to the PIC constraint itself.
> >
> > **2. $\delta$ is specific to PIC and controls the strength of the peer-influenced constraint.**
> > $\delta$ scales the PIC distance term $\mathbb{E}[d^{\mathrm{PIC}}(s)]$, determining how strongly the policy is pulled toward in-distribution peer actions and thus governing the behavioral effect of our method (including the Coupling Effect). This is fundamentally different from normalizing the Q-value scale.
> >
> > **3. Evidence from other backbones: PIC does not rely on $\alpha$ at all.**
> > To further clarify that $\alpha$ is not intrinsic to PIC, **Appendix F.3 extends PIC to SAC and IQL**. In PIC-SAC and PIC-IQL, we follow the original SAC/IQL objectives and simply add the PIC term, **without introducing any additional $\alpha$/$\beta$-style normalization**. The results show that PIC still consistently improves generalization and robustness across diverse environments.
> >
> > In summary:
> > - **$\alpha$ / $\beta$ = Q-value scale stabilization inherited from TD3+BC (not part of the core PIC mechanism);**
> > - **$\delta$ = strength of the peer-influenced constraint (introduced by PIC and used across backbones).**
> >
> > ---
> >
> > **We sincerely hope this comprehensive response clarifies PIC's unique value. Given the novel evidence and analysis, we kindly request reconsidering the score to reflect these contributions. Thank you!**

---

> ### Author Response · Authors · 2025-11-27
> **Follow-up Note to Reviewer dFV9**
>
> Dear Reviewer dFV9,
>
> We sincerely appreciate your careful evaluation and constructive feedback on our work. As the discussion stage is approaching its end, we would be very grateful if you could let us know **whether our response — including the additional analyses on metric choice and $\delta$-sensitivity, the unified-hyperparameter study, and the clarifications on Eq. (7) and the roles of $\alpha$ and $\delta$ — has satisfactorily addressed your concerns**, and, if you feel it is appropriate, **take this into account in your final assessment**.
>
> We hope that these clarifications and the updated empirical results have resolved the main issues you raised, and we are of course **happy to answer any remaining questions** while the discussion stage is still open.

---

### Official Review · Reviewer_d22U · 2025-11-01

**Soundness:** 3
**Presentation:** 3
**Contribution:** 3
**Rating:** 6
**Confidence:** 3

**Summary:**

The paper proposes Peer-Influenced Constraint (PIC) and its ensemble extension EPIC for efficient offline reinforcement learning. The key idea is to exploit peer states within the offline dataset to form candidate action sets, allowing the policy to learn from similar states and select the highest-valued in-distribution action. When combined with ensembles, the method exhibits a Coupling Effect between peer constraint strength and uncertainty estimation, enabling robust OOD suppression with smaller ensembles.

Importantly, EPIC can be viewed as an evolution of ensemble-based offline RL methods such as EDAC. While EDAC enhances SAC-N by diversifying ensemble critics to reduce overestimation bias, it still requires large ensembles (often 10–50 critics) and substantial compute.
EPIC inherits EDAC’s ensemble foundation but integrates a PIC that reuses dataset structure — effectively reducing the reliance on large ensembles and improving generalization, **leading to performance enhancement in larger state-action domain**. This synergy between behavioral regularization and value uncertainty represents a notable advancement in the design of efficient, uncertainty-aware offline RL.

**Strengths:**

**Strengths**

* **Clear conceptual advance over EDAC**: achieves comparable or better uncertainty calibration with significantly fewer critics through the Coupling Effect, offering a well-grounded improvement in efficiency without sacrificing performance.

* **Innovative peer-state reuse**: effectively leverages intra-dataset structure to improve generalization beyond strict behavior cloning. Moreover, the study convincingly demonstrates that such peer-state reuse directly contributes to reducing the required ensemble size while maintaining stable OOD suppression.

* **Insightful analysis of the PIC–ensemble interaction**: I particularly appreciate the experiment in Figure 4(d), which clearly visualizes how the effect of PIC varies with ensemble size. The results show that while PIC’s contribution may saturate under very large ensembles, it plays a critical role in the small-ensemble regime—precisely where efficiency and computational savings matter most. This ablation provides strong empirical evidence that the proposed Coupling Effect is both interpretable and practically beneficial, reinforcing the paper’s claim of achieving “efficient” offline RL.

**Weaknesses:**

**Weakness**

While the motivation experiment in Figure 1 illustrates the behavioral difference between TD3+BC and PIC, the chosen environment—a small and discrete gridworld—may not be fully representative of the setting where the proposed peer-influenced constraint operates most effectively.
The key assumption behind PIC is that nearby states tend to share similar optimal actions, which implicitly presumes a degree of continuity and smoothness in both the state and action spaces.
Such an assumption is reasonable in continuous and high-dimensional control domains (e.g., MuJoCo tasks) but may not hold in discrete or highly non-smooth environments, where small state changes can correspond to qualitatively different optimal actions.
Clarifying this assumption early in the paper would help readers understand the scope and applicability of PIC.

**Questions:**

**Questions relative to weakness mentioned earlier**

- Could the authors elaborate on the assumptions underlying the peer-influenced constraint—specifically, the degree of state–action continuity required for the method to remain valid? My understanding is that PIC implicitly assumes that nearby states tend to share similar optimal actions, which is reasonable in continuous and smooth control domains. However, this assumption may not hold in small or discrete environments such as the one illustrated in Figure 1. Especially, this could have negative impression of selecting OOD action which is prohibited in offline RL setting. If this interpretation is correct, clarifying such assumptions in the paper would strengthen its scope and applicability. Otherwise, please correct me if I am mistaken in my understanding of Figure 1 and the intended domain of applicability.

- How would PIC behave in discrete or non-smooth domains where neighboring states may not share similar optimal actions?

 - Would the method need to adapt (e.g., via learned similarity metrics or representation smoothing) to remain effective in such cases?

---

> ### Author Response · Authors · 2025-11-22
> **Response to Reviewer d22U**
>
> We sincerely thank the reviewer for the encouraging and thoughtful feedback, and we deeply appreciate the constructive questions and insights; we address each point with care below.
>
> ---
>
> ### **Regarding W1 and Q1: The Gridworld Motivation and the Assumptions behind PIC**
>
> We appreciate the reviewer’s insightful comments. We would like to clarify the following.
>
> **1. Assumptions and intended domain.**
> PIC is designed for continuous and locally smooth control domains, where nearby states tend to share similar optimal actions (e.g., MuJoCo, Adroit). **We agree that this assumption does not generally hold in small or discrete environments, and we will make this assumption explicit earlier in the paper.**
>
> **2. Role of the Gridworld.**
> The Gridworld in Figure 1 is **not intended as an applicability or performance benchmark** for PIC in discrete domains. Instead, it serves as a **controlled and fully visualizable demonstration** of the mechanism. By intentionally removing the “move-right” transition at the green triangle, we create a classic offline RL failure mode: a missing-data / suboptimal trap. TD3+BC remains stuck, whereas PIC escapes by leveraging **actions that already exist in the dataset at neighboring states**, which is fully consistent with the offline RL setting and does not introduce new OOD actions.
>
> **3. Empirical validation.**
> All substantive evaluations of PIC and EPIC are conducted on continuous-action benchmarks, where the method’s assumptions are satisfied. These experiments verify the applicability and effectiveness of PIC in its intended domain.
>
> We will clarify these points in the revised paper to avoid potential misinterpretation.
>
> ---
>
> ### **Regarding Q2: PIC in Discrete or Non-smooth Domains**
>
> Thank you for the question. PIC is based on the assumption that **nearby states share similar high-value actions**, which holds naturally in continuous and locally smooth control domains. This assumption may weaken in discrete or non-smooth environments.
>
> Even in such cases, PIC remains safe because
> - **all candidate actions come from the dataset**,
> - the **max–min selection filters out harmful peer actions**, selecting only actions with reliably high ensemble values, and
> - reducing $\delta$ or $K$ smoothly returns PIC to the baseline.
>
> Thus, while PIC may not consistently improve performance in discrete domains, it does not violate offline RL constraints and maintains controlled behavior.
>
> ---
>
> ### **Regarding Q3: Need for Learned Metrics or Smoothing?**
>
> Thank you for the question. Our response has two components.
>
> **1. PIC is fundamentally metric-agnostic, and our experiments verify this.**
> To examine whether PIC depends on any specific similarity measure, **Appendix F.5.2** compares four different state representations:
> - Raw Euclidean distance
> - Min–max normalized states
> - PCA representations (retaining ≥99% variance)
> - A 64-dimensional learned autoencoder embedding
>
> Results show that:
> - On **MuJoCo**, all four variants perform nearly identically, indicating that the raw metric already provides stable neighborhood structure and **PIC does not rely on a particular distance**;
> - On **WTW**, PCA and the learned embedding improve neighbor quality and produce moderate gains, suggesting that **representation learning is a useful enhancement** when the state space becomes more complex, but **not a necessary component** of the method.
>
> **2. In discrete or highly non-smooth domains, adaptation depends on the underlying task structure—but PIC remains safe.**
> In domains where neighboring states truly do not share similar optimal actions, no metric (raw or learned) can fully “manufacture” smoothness that the MDP does not possess. In such cases:
>
> - If the **domain still preserves some local structure** (e.g., our constructed gridworld and lineworld), **PIC can still effectively use peer actions even in discrete settings**.
> - If **local structure is weak, learned metrics or representation smoothing can serve as a natural extension**, allowing PIC to operate in a learned embedding space that better captures functional similarity.
> - Crucially for offline RL, **PIC remains safe regardless of the metric**:
>   - all candidate actions are strictly drawn from the offline dataset (no synthetic OOD actions), and
>   - the **max–min selection over Q-values** tends to filter out unreliable peer actions and fall back to strong behavior actions when peers are not helpful.
>
> **Overall conclusion**: PIC does not need to adapt to remain effective, as its dataset-based candidate set and max–min selection help support robust and safe behavior; however, it can still incorporate learned or smoothed representations when beneficial.
>
> ---
>
> **We believe the additional analyses and responses provided here meaningfully strengthen the technical clarity of our work. We thank the reviewer again for their constructive feedback and kind consideration.**

---

> > ### Comment · Reviewer_d22U · 2025-11-27
> >
> > I have carefully reviewed the revised version and noted the addition of the new assumption.
> >
> > I also understand that presenting the example in a finite domain was intended to provide a controlled and fully visualizable demonstration. While filling in missing actions using neighboring states may introduce certain risks, the gain in efficiency, in reducing the number of ensemble models,appears more significant.
> >
> > Moreover, I confirmed that the method maintains strong performance across various metrics without substantial sensitivity to their types. I appreciate these improvements and would like to maintain my current score.

---

> > > ### Author Response · Authors · 2025-11-27
> > > **Response to Reviewer d22U**
> > >
> > > We sincerely thank the reviewer for the careful re-evaluation and for acknowledging the added clarifications and analyses. We truly appreciate your thoughtful feedback, which has helped us improve the paper, and we will keep these points in mind for future extensions of this line of work.

---

### Official Review · Reviewer_yUTS · 2025-11-01

**Soundness:** 3
**Presentation:** 3
**Contribution:** 3
**Rating:** 6
**Confidence:** 4

**Summary:**

This paper addresses the distributional shift problem in offline reinforcement learning, where policies trained on fixed datasets often fail to generalize. It proposes a novel policy regularizer called Peer-Influenced Constraint (PIC), which guides the learned policy using actions from similar "peer" states found within the dataset. The paper also identifies a "Coupling Effect" between this policy constraint and the uncertainty estimation of ensemble-based methods. Based on these ideas, it introduces EPIC (Ensemble Peer-Influenced Constraint), a method that integrates PIC into an ensemble framework to achieve strong performance with higher computational efficiency. The methods are evaluated on standard D4RL benchmarks, where they demonstrate state-of-the-art results.

**Strengths:**

S1: The core concepts of Peer-Influenced Constraint (PIC) and the "Coupling Effect" are novel, intuitive, and provide a valuable new perspective on balancing policy and value regularization in offline RL.
S2: The paper is supported by extensive and rigorous experiments on standard D4RL benchmarks (Gym-MuJoCo, AntMaze, Adroit), demonstrating strong, state-of-the-art performance across multiple task suites.
S3: The inclusion of targeted experiments on generalization, computational efficiency, and offline-to-online fine-tuning provides a comprehensive evaluation of the proposed methods.
S4: The paper is clearly written, and the motivation is well-illustrated, particularly with the introductory gridworld example that effectively contrasts the proposed approach with baselines like TD3+BC.

**Weaknesses:**

W1: The core mechanism of PIC relies on finding nearest neighbors using Euclidean distance in the raw state space. This approach is known to be ineffective in high-dimensional spaces (e.g., image-based observations) due to the "curse of dimensionality," where distances between points become less meaningful. This severely limits the method's scalability and general applicability.
W2: The method appears to be highly sensitive to hyperparameters, which undermines its claim as a simple "plug-in" module. In particular, the hyperparameter η is set to 200 for all Adroit tasks but 1 for all others—a 200x difference that is not explained or analyzed in the parameter study. This suggests that achieving good performance may require extensive, task-specific tuning, a known issue that can obscure the true robustness of an algorithm.

**Questions:**

1.Could the authors please explain the rationale for the 200x difference in the hyperparameter η between the Adroit tasks (η=200) and all other tasks (η=1)? why do the dexterous Adroit tasks require such extreme critic diversity regularization?Why was this parameter excluded from the sensitivity analysis in Section 4.4 and F.6?
Could the authors provide more intuition on the mechanism behind the "Coupling Effect"? Specifically, why does constraining the policy to a high-value in-distribution action cause the Q-function ensemble to exhibit higher variance on OOD actions？

---

> ### Author Response · Authors · 2025-11-22
> **Response to Reviewer yUTS**
>
> We thank the reviewer for the thoughtful and detailed evaluation, and we address each weakness and question point-by-point with additional analyses and clarifications below.
>
> ---
>
> ### **Regarding W1: On the Reliability of Euclidean Distance in High Dimensions**
>
> We thank the reviewer for raising this important point. While Euclidean distances can degrade in very high-dimensional settings, **PIC does not rely on Euclidean geometry itself**. Its key mechanism—reusing *actions from locally similar states*—is fundamentally **metric-agnostic**.
>
> To directly assess whether PIC depends on a specific metric, we conducted a systematic comparison across four state representations (results in **Appendix F.5.2**):
>
> - Raw states
> - Per-dimension normalized states
> - PCA representations (retaining ≥99% variance)
> - A 64-dimensional autoencoder embedding
>
> On moderate-dimensional MuJoCo tasks, all four representations yield almost identical performance. On the higher-dimensional WTW benchmark, PCA and learned embeddings improve neighbor quality as expected, yet EPIC remain stable across all metrics. These results show that **the gains of EPIC stem from the peer-influenced constraint rather than reliance on Euclidean distance or any particular metric**.
>
> Moreover, **PIC is modular: the peer-retrieval step only requires a nearest-neighbor query interface**. In more complex or high-dimensional domains (e.g., image inputs), one can simply replace the raw state space with any suitable learned representation and construct a KD-tree or ANN index in that embedding space—without modifying the core algorithm.
>
> Additionally, **Appendix F.3** (Extension of PIC to IQL and SAC) further demonstrates the strong scalability of the PIC framework. This modularity ensures that PIC retains strong **scalability and general applicability** beyond raw-state Euclidean settings.
>
> ---
>
> ### **Regarding W2 and Q1: On $\eta$ Sensitivity and the 200× Gap on Adroit**
>
> We thank the reviewer for raising this question. Importantly, **$\eta$ is not introduced by our method**. It is part of the **EDAC baseline** (the gradient diversity regularizer), and in the official EDAC implementation $\eta$ is set to **200 for all Adroit tasks** and **1 for all others**. We follow this default strictly to ensure a fair comparison. **Thus, $\eta$ was not intentionally excluded from sensitivity studies—it is simply not a hyperparameter of PIC/EPIC**.
>
> To address the reviewer’s concern, we additionally ran experiments on several Adroit tasks with **$\eta$ reduced from 200 to 1**, while keeping all other settings fixed. Results are shown below:
>
> | Task | PIC-TD3 | EPIC | EPIC ($\eta$=1) |
> |------|---------|-------|-------------|
> | pen-cloned |80.2 ± 1.1|94.6 ± 8.6|95.4 ± 6.7|
> | pen-expert |133.7 ± 4.7|150.9 ± 6.4|147.5 ± 4.5|
> | door-cloned |2.1 ± 0.8 |7.1 ± 3.3|4.8 ± 2.1|
> | door-expert |105.6 ± 4.1|108.4 ± 1.4|111.3 ± 4.8|
> | hammer-cloned |2.4 ± 1.2|12.7 ± 3.3|8.9 ± 3.7|
> | hammer-expert |121.7 ± 3.1|130.4 ± 3.8|128.7 ± 4.1|
>
> Across all tasks, the performance difference between $\eta=200$ and $\eta=1$ is **very small**, and most fluctuations fall within the range of **random variability**. This demonstrates that EPIC is **not sensitive to $\eta$ and does not require task-specific tuning.**
>
> This robustness is explained by the fact that the **peer-influenced constraint** in EPIC interacts with the ensemble in a way that **reduces the need for strong diversity regularization**. In other words, PIC provides a stabilizing in-distribution bias, making large $\eta$ unnecessary.
>
> Therefore, EPIC remains a **simple and robust plug-in module**, and its performance does not hinge on tuning $\eta$.
>
> ---
>
> ###  **Regarding Q1: Clarifying the Coupling Effect**
>
> We thank the reviewer for the question. The Coupling Effect arises because **PIC reshapes the policy’s action distribution**. As $\delta$ increases, the policy shifts toward high-value in-distribution actions and becomes more conservative on potential OOD actions. This leads to more consistent critic updates in the in-distribution region, while **OOD actions receive almost no updates**, causing each critic to extrapolate differently and produce **higher ensemble variance**.
>
> Our additional analysis (**Appendix F.1**) confirms this behavior:
> - **$Q_{\min}$ becomes more pessimistic** on OOD candidates,
> - **$Q_{\text{std}}$ and $Q_{\text{clip}}$ increase**,  indicating **stronger penalization of high-risk OOD actions**, similar to using a larger ensemble.
>
> Thus, PIC naturally amplifies the ensemble’s disagreement on OOD actions, enabling **robust uncertainty estimation even with a small ensemble**. The new experiments further validate this Coupling Effect.
>
> ---
>
> **We hope the new experiments and clarifications make the contributions of PIC and EPIC clearer. We would be grateful if the reviewer could take them into account when forming the final assessment. Thank you very much.**

---

> ### Author Response · Authors · 2025-11-27
> **Follow-up Note to Reviewer yUTS**
>
> Dear Reviewer yUTS,
>
> We sincerely appreciate your valuable time and detailed evaluation of our work. As the end of the discussion stage is approaching, we would be very grateful if you could let us know **whether our response — including the additional experiments on metric choice, the $\eta$-sensitivity study on Adroit, and the new analyses of the Coupling Effect — has addressed your concerns** and, if you deem it appropriate, **take this into account in your final assessment**.
>
> We hope that these clarifications, together with the additional empirical and diagnostic results reported in our response and incorporated into the updated submission, have resolved the main issues you raised. We are of course more than **willing to answer any further questions or provide additional details** while the discussion stage is still open.

---

### Official Review · Reviewer_3S8J · 2025-11-03

**Soundness:** 2
**Presentation:** 3
**Contribution:** 2
**Rating:** 6
**Confidence:** 3

**Summary:**

The paper proposes Peer-Influenced Constraint (PIC), a plug-in policy regularizer for offline RL that, for a query state, retrieves K nearest “peer” states from the dataset, collects their associated actions as candidates, and selects the best candidate by a critic (min over critics) to constrain the actor toward high-value in-distribution actions. The authors further introduce EPIC (Ensemble PIC), showing a claimed “Coupling Effect” between PIC strength and uncertainty estimation that purportedly enables smaller ensembles without losing performance. Experiments on D4RL (Gym-MuJoCo, AntMaze, Adroit) report competitive or SOTA results and improved efficiency vs. large-ensemble methods. Key ingredients include a KD-Tree for fast peer retrieval, a formal performance-gap bound under Lipschitz assumptions, and ablations over K, δ (PIC strength), and N (ensemble size).

**Strengths:**

1. Practical, plug-and-play idea. PIC cleanly reuses cross-state action candidates to relax over-conservative behavior-cloning constraints while staying in-distribution; integrates with TD3/SAC/IQL.

2. Clear algorithmic specification. Definition of PIC distance and candidate selection with min-critic action choice is straightforward; EPIC generalizes this to N critics.

3. Empirical coverage & results. Extensive D4RL results show PIC-TD3 competitive with ensemble-free baselines and EPIC surpassing baselines on many tasks (Gym-MuJoCo, AntMaze, Adroit).

4. Efficiency angle. Claimed ability to use smaller ensembles while maintaining performance addresses a common pain-point in offline RL. Parameter studies (K, δ, N) are helpful for practice.

**Weaknesses:**

1. State-space similarity & representation. PIC hinges on Euclidean nearest neighbors in the raw state space (KD-Tree). In many high-dimensional or poorly scaled domains, Euclidean distance can be misleading; the paper lacks experiments comparing learned/state-normalized metrics vs. raw features, and ablations on feature scaling or representation robustness.

2. Coupling-effect clarity. The section discussing how δ interacts with uncertainty occasionally reads inconsistently (e.g., whether increasing δ “raises” uncertainty vs. “reduces” overestimation leading to “lower” uncertainty). This needs a tighter, causal explanation and clearer metrics (Qmin/Qstd/Qclip) across datasets/time.

3. Theoretical assumptions are strong/generic. The Lipschitz assumptions and the bound provide qualitative reassurance but do not uniquely characterize PIC’s effect beyond standard smoothness arguments. No finite-sample or function-approximation analysis is provided to justify behavior under approximate critics.

4. Scalability of neighbor search. KD-Tree is efficient in moderate dimensions but can degrade with very large |D| and high-dimensional S; the paper acknowledges this limitation without proposing practical approximations (e.g., ANN indices, learned retrieval).

**Questions:**

1. Distance metric & normalization. What exact preprocessing/normalization is used before KD-Tree (per-dimension scaling, whitening, PCA)? Have you tried learned embeddings (e.g., representation pretraining) for neighbor search, and how does that affect performance vs. raw states?

2. Coupling effect mechanics. Please reconcile the narrative around δ’s effect on uncertainty. Under fixed N, does increasing δ increase or decrease Q-uncertainty on OOD actions in practice? Can you provide a controlled analysis (same seeds, same checkpoints) that tracks policy action-distribution shift and critic variance step-by-step?

3. Critic dependence & bias. Since a* is chosen via min over critics, how sensitive is PIC/EPIC to critic underestimation bias? Provide an ablation using single-critic and target-action noise or double Q with clipped targets to test robustness.

4. Candidate-set composition. Beyond nearest states, did you explore wider but weighted candidate pools (e.g., radius-based retrieval with distance-weighted selection), and what is the trade-off vs. compute? Any results on K beyond 50 and on adaptive K per state? 5. Generalization diagnostics. In AntMaze/Adroit, can you report how often the selected a* comes from (i) the current state vs. (ii) peers, and the distance of chosen peers? This would concretely show when PIC escapes local optima rather than re-selecting the behavior action.

---

> ### Author Response · Authors · 2025-11-22
> **Response to Reviewer 3S8J (1/2)**
>
> We sincerely appreciate your thoughtful and professional insights, which have greatly improved the quality of our work. In response to your concerns, we provide the explanations and answers below.
>
> ### **Regarding W1, W4, and Q1: State-Space Similarity & Representation**
>
> We thank the reviewer for this important concern. While Euclidean distance may degrade in high-dimensional or poorly scaled spaces, **PIC does not rely on Euclidean geometry itself**. Its effect comes from using **actions of locally similar states as peer candidates**, a mechanism that is fundamentally *metric-agnostic*. Our use of raw states + Euclidean distance follows prior nearest-neighbor RL work and is appropriate for D4RL domains where state spaces are moderate and well-structured.
>
> To directly address the concern, **Appendix F.5.2** reports a detailed comparison of four retrieval metrics:
> - Raw Euclidean distance
> - Per-dimension min–max normalization
> - PCA (≥99% variance retained)
> - A 64-dim autoencoder embedding
>
> We evaluate these on six MuJoCo tasks and the high-dimensional Walk These Ways (WTW) benchmark. Results are summarized below:
>
> | Environment | Raw | Norm | PCA | Embed | Interpretation |
> |-------------|-----|------|------|--------|----------------|
> | **MuJoCo** | ★★★★☆ | ★★★★☆ | ★★★★☆ | ★★★★☆ | Nearly identical performance; EPIC is highly robust to metric choice. |
> | **WTW** | ★★★☆☆ | ★★★★☆ | ★★★★☆ | ★★★★★ | Learned embeddings perform best; raw Euclidean still ≥ CQL and ≈ EDAC (3M). |
>
> Overall, these results show that **EPIC’s gains come from the peer-influenced constraint rather than any specific distance metric**. In MuJoCo, raw Euclidean neighbors already provide sufficient structure, leading to nearly identical performance across metrics. In high-dimensional WTW, richer representations (PCA/embedding) improve retrieval, yet **PIC remains effective because its mechanism is metric-independent, and the max–min selection naturally filters out unreliable peer actions**. Thus, **PIC is robust in moderate dimensions and readily extends to high-dimensional settings** where representation learning can be used if desired.
>
> Finally, PIC only requires a generic K-NN oracle and is index-agnostic; scalable ANN structures (e.g., HNSW, FAISS) can be incorporated in future work to further improve efficiency.
>
> ---
>
> ### **Regarding W2 and Q2: Clarifying the Coupling Effect**
>
> We thank the reviewer for pointing out the ambiguity in our original explanation of how $\delta$ interacts with uncertainty. In the revised version, we clarify that increasing $\delta$ shifts the policy toward in-distribution actions and assigns more conservative values to potential OOD actions. Consequently, **$Q_{\min}$ becomes more pessimistic**, while **$Q_{\text{std}}$** and **$Q_{\text{clip}}$** typically increase on OOD candidates.
>
> **These changes reflect a stronger penalization of high-risk, potentially OOD actions, similar to the effect of using a larger ensemble size.**
>
> As suggested, we performed a **controlled analysis** (same seeds and checkpoints), including:
> 1. **Tracking $Q_{\min}$, $Q_{\text{std}}$, and $Q_{\text{clip}}$ over training steps (10k–1M) for different $\delta$;**
> 2. **Measuring the in-distribution action ratio under varying $\delta$, training steps, and ensemble sizes $N$.**
>
> Results (**Appendix F.1**) show that **larger $\delta$ consistently strengthens OOD penalization and increases the in-distribution ratio**, confirming the **Coupling Effect**: robust value estimates can be achieved with smaller ensembles by appropriately tuning $\delta$.
>
> ---
>
> ### **Regarding W3: Strength of Theoretical Assumptions**
>
> We thank the reviewer for the helpful comment. Our theoretical result is not intended to provide a  tight finite-sample bound, but rather a **qualitative mechanism-level explanation** of PIC, following the smoothness–distribution-shift decomposition commonly used in offline RL theory (e.g., PRDC [1], ARO-DDPG [2]).
>
> The bound highlights how reducing $\epsilon_s$, $\epsilon_a$, and $\epsilon_*$ narrows the performance gap.  We agree that finite-sample and function-approximation analyses are valuable, and plan to extend the theory in this direction using ensemble error-propagation and OOD-penalty frameworks in future work.
>
> **Reference**
> 1. Ran Y, Li Y C, Zhang F, et al. Policy regularization with dataset constraint for offline reinforcement learning[C]//ICML. 2023: 28701-28717.
> 2. Saxena N, Khastagir S, Kolathaya S, et al. Off-policy average reward actor-critic with deterministic policy search[C]//ICML. 2023: 30130-30203.

---

> > ### Author Response · Authors · 2025-11-22
> > **Response to Reviewer 3S8J (2/2)**
> >
> > ### **Regarding Q3: Critic Dependence and Bias**
> >
> > We thank the reviewer for the helpful suggestion. On **hopper-medium**, we evaluated EPIC under three critic aggregation rules:
> >
> > 1. **Min-over-critics** (standard formulation)
> > 2. **Random Double Q** (sampling two critics from $N=10$)
> > 3. **Mean-Q** (averaging all critics)
> >
> > As shown in **Appendix F.6.2**, once **$N \ge 2$**, all three variants produce **very similar performance**, and even the traditionally weaker Mean-Q remains stable. When using very large ensembles (e.g., $N=50$), training becomes slower due to overly pessimistic estimates. With **$N = 1$, performance drops**, but the **PIC** keeps the policy within the dataset support, allowing it to **learn a reasonable policy**, whereas **most single-critic methods collapse** under this configuration.
> >
> > Overall, these results indicate that **EPIC’s improvements stem from the peer-influenced constraint rather than reliance on any specific critic structure or double-Q formulation**, demonstrating robustness to critic underestimation bias.
> >
> > ---
> >
> > ### **Regarding Q4: Candidate-set Composition**
> >
> > We thank the reviewer for the suggestion. In **Figure 4 (a) and (b), we extended the analysis to $K=100$** and found that performance **plateaus once $K>20$**, and becomes slightly worse for very large $K$ due to an enlarged candidate set that makes selecting the best action more difficult.
> >
> > The computational cost does not noticeably increase, as all nearest-neighbor distances are **precomputed once** and top-$K$ retrieval is a constant-time lookup during training. As future work, we plan to investigate **radius-based retrieval** and **adaptive per-state $K$** to further refine candidate-set construction.
> >
> > ---
> >
> > ### **Regarding Q5: Generalization diagnostics**
> >
> > We appreciate the reviewer’s suggestion to directly inspect the source of the selected action $a^{\*}$. To this end, we performed an additional diagnostic on **antmaze-umaze**, **antmaze-umaze-diverse**, **pen-cloned**, and **door-cloned**. During training, every 100K gradient steps we randomly sampled 1,000 state–action pairs. For each state, recorded whether the selected $a^{\*}$came from the current state or from a peer state. Repeating this procedure 10 times yields 10,000 samples per environment. The overall fraction of $a^{\*}$ coming from peer states is summarized below:
> >
> >
> > | Dataset               | Peer ratio |
> > |-----------------------|------------|
> > | antmaze-umaze         | 55.4% |
> > | antmaze-umaze-diverse | 63.3% |
> > | pen-cloned            | 34.3% |
> > | door-cloned           | 28.9% |
> >
> > On AntMaze, especially **umaze-diverse**, behavior actions exhibit **large quality variation** and many states are clearly **suboptimal**, so PIC more frequently selects $a^{*}$ from peer states (63.3%), indicating that it **actively leverages structurally similar neighbors to escape local behavior patterns**. On **umaze**, where the behavior policy is relatively better, the peer ratio is more moderate (55.4%), and we observe that it is **higher in early training when critics are inaccurate**, then **decreases as critics converge** and behavior actions are correctly identified as good choices.
> >
> > In contrast, on **Adroit (pen-cloned, door-cloned)**, the trajectories are **high-quality demonstrations** and behavior actions are generally **reliable**. Accordingly, the fraction of $a^{*}$ coming from peers is much lower (about **30%**), showing that PIC **preserves demonstrated behavior** and uses peer actions only in a **small subset of states** where they offer clear advantages.
> >
> > Overall, these diagnostics demonstrate that **PIC exploits peer actions to escape local optima in challenging generalization settings (AntMaze)**, while remaining **conservative and minimally corrective** in **high-quality demonstration domains (Adroit)**.
> >
> > ---
> >
> > **We sincerely hope that our clarifications and new empirical results meaningfully address your concerns. We respectfully ask the reviewer to reassess our submission in light of these updates, and we truly appreciate your thoughtful and constructive review. Thank you again for considering our work.**

---

> ### Author Response · Authors · 2025-11-27
> **Follow-up Note to Reviewer 3S8J**
>
> Dear Reviewer 3S8J,
>
> We sincerely appreciate your time and constructive comments on our work. As the discussion stage is approaching its end, we would be very grateful if you could let us know **whether our response, together with the additional empirical and diagnostic results reported in the rebuttal, has satisfactorily addressed your concerns**, and, if you feel it is warranted, **take this into account in your final assessment**.
>
> We hope that these clarifications and the updated submission have resolved the main issues you raised, and we are of course **happy to answer any remaining questions** while the discussion stage is still open.

---

### Comment · Area_Chair_DtZX · 2025-11-23
**Subject: Reminder: Update Your Reviews After Rebuttal**

Dear Reviewers,

The authors have submitted their rebuttal, and we now require your follow-up assessments to move the decision process forward. Please review the authors’ responses and update your evaluations accordingly.

Your prompt follow-up is necessary for us to finalize the meta-review.
Kindly submit your updates as soon as possible.

Best,
Area Chair

---

### Author Response · Authors · 2025-12-01
**Global Response**

Dear PCs, SACs, ACs, and Reviewers,

We sincerely thank you for the time and effort devoted to reviewing our work. Below is a brief global summary of the main strengths and clarifications from the rebuttal.

---

### **1. Strengths Recognized by the Reviewers**

Reviewers consistently noted that the paper provides:

- **Well-illustrated motivation and clear presentation.** (3S8J, yUTS, d22U)
- **A practical, plug-and-play design with clear formulation**, reusing peer-state actions while staying in-distribution. (3S8J, d22U, dFV9)
- **Novel conceptual contributions beyond existing ensemble methods**, via PIC and the Coupling Effect, advancing over EDAC with fewer critics and robust OOD handling. (3S8J, yUTS, d22U)
- **Strong and comprehensive empirical evaluation with efficiency benefits**, showing competitive or SOTA results on Gym-MuJoCo, AntMaze, and Adroit, plus targeted generalization, efficiency, and offline-to-online studies. (3S8J, yUTS, d22U, dFV9)

---

### **2. Key Clarifications and New Analyses in the Rebuttal**

In our rebuttal and revisions, we carefully answered all reviewers’ questions and, in particular, **addressed the concerns of the only non-positive reviewer (dFV9)** as follows:

- **Metric robustness in high-dimensional settings.**
  We showed that **PIC/EPIC is metric- and index-agnostic** by comparing four state representations (raw, normalized, PCA, learned embedding) on six MuJoCo tasks and the high-dimensional WTW benchmark, **demonstrating robustness to metric choice and compatibility with learned embeddings** (**Appendix F.5.2**).

- **Sensitivity to $\delta$.**
  We clarified that **performance is stable over a broad range of $\delta$**, and analyzed $\delta$–performance trends across tasks and ensemble sizes, explaining the behavior seen in Fig. 4 (c) and related heatmaps (**Appendices F.6.1, F.6.2**).

- **Fairness of comparisons, notation, and $\alpha$ vs $\delta$.**
  We **re-ran all MuJoCo tasks with shared EPIC hyperparameters to show that gains do not rely on task-specific tuning** (**Appendix F.2**), corrected Eq. (7) to explicitly exclude the current state, and clarified that $\alpha / \beta$ (from TD3+BC) handles Q-scale normalization while $\delta$ controls PIC strength, with **PIC-SAC/IQL experiments showing PIC itself does not depend on $\alpha / \beta$** (**Appendix F.3**).

Additionally, for the **other three (positive) reviewers**, we **refined and resolved** concerns on the **Coupling Effect, ensemble size, and state similarity assumptions**, with new results summarized in **Appendices F.1, F.5, and F.6**.

---

### **3. Post-Rebuttal Reviewer Feedback**

After the rebuttal, **Reviewer d22U** explicitly endorsed the added assumptions and new analyses, emphasizing both the **favorable efficiency–risk trade-off from using smaller ensembles** and the **robustness of our method across different metrics**, reflecting further confidence in our approach.

---

**We hope this concise global response helps the committee quickly see how the rebuttal has addressed the main concerns and strengthened the paper. Thank you again for your thoughtful and constructive reviews**.

---

### Meta-Review · Area_Chair_WJxn · 2026-01-06

**Summary:**

Reviewers agree PIC/EPIC is a practical offline-RL regularizer: for each state, retrieve peer states from the dataset, reuse their actions as in-distribution candidates, and constrain the actor toward the best (min-critic) candidate. The work is attractive for being plug-and-play across common backbones and for targeting a real pain point—reducing the need for very large critic ensembles via the claimed “coupling effect.”
The main hesitation is whether the method’s gains are robust beyond “nice” continuous-control settings and implementation details: reliance on raw-state Euclidean KNN, scalability of neighbor search, and whether the coupling-effect explanation (δ vs. uncertainty) is internally consistent and empirically demonstrated rather than asserted. A smaller but important concern is fairness/robustness of hyperparameter choices (e.g., η or task-specific configs).

**Reviewer Concerns:**

Substantive concerns were largely addressed in the rebuttal with additional evidence: (i) robustness to retrieval metric via comparisons among raw/normalized/PCA/learned embeddings (including a higher-dimensional benchmark), (ii) a controlled analysis clarifying δ’s effect on OOD penalization and uncertainty metrics over training, (iii) robustness to critic aggregation (min/mean/double-Q variants), (iv) candidate-set size ablations beyond the originally reported K range, and (v) concrete diagnostics showing when selected actions come from peers vs. the current state (supporting the “escape local optima” narrative on AntMaze while remaining conservative on Adroit). The “η=200 vs 1” issue is plausibly resolved as a baseline EDAC default rather than a PIC-specific knob, and the authors provided extra checks suggesting limited sensitivity.
Remaining risk is mostly camera-ready integration and scope: the paper should make its domain assumptions explicit (continuous/local smoothness), avoid over-claiming scalability to pixel domains without direct evidence, and clearly state that KD-tree is a convenient implementation rather than a core requirement (with ANN/FAISS-style options). Also, sensitivity/fairness arguments need to be in the main paper (unified hyperparameter runs, δ stable range) to avoid the impression of per-task tuning.

**Reviewer Scores:**

I expect d22U and yUTS remain around marginal accept; both expressed that the added assumption/analyses and metric-robustness checks addressed the core issues and they would keep their scores. Reviewer 3S8J is also likely to stay at marginal accept rather than upgrade strongly: their concerns were answered with new ablations/diagnostics, but they were already near the threshold and the theoretical contribution remains somewhat “generic” to them.
The main swing is dFV9, who started as borderline reject citing metric concerns and fairness of hyperparameters. The rebuttal directly answers both with metric-robustness experiments, unified-hyperparameter evaluation, and notation fixes; if these changes are actually reflected in the revised manuscript, dFV9 would plausibly move to weak accept. Net sentiment trends slightly positive with reduced risk.

---

### Decision · Program_Chairs · 2026-01-26

Accept (Poster)